# ARE NEURAL NETS MODULAR? INSPECTING FUNCTIONAL MODULARITY THROUGH DIFFERENTIABLE WEIGHT MASKS

**Róbert Csordás**
IDSIA / USI / SUPSI
robert@idsia.ch

**Sjoerd van Steenkiste**
IDSIA / USI / SUPSI
sjoerd@idsia.ch

**Jürgen Schmidhuber**
IDSIA / USI / SUPSI / NNAISENSE
juergen@idsia.ch

## ABSTRACT

Neural networks (NNs) whose subnetworks implement reusable functions are expected to offer numerous advantages, including compositionality through efficient recombination of functional building blocks, interpretability, preventing catastrophic interference, etc. Understanding if and how NNs are modular could provide insights into how to improve them. Current inspection methods, however, fail to link modules to their functionality. In this paper, we present a novel method based on learning binary weight masks to identify individual weights and subnets responsible for specific functions. Using this powerful tool, we contribute an extensive study of emerging modularity in NNs that covers several standard architectures and datasets. We demonstrate how common NNs fail to reuse submodules and offer new insights into the related issue of systematic generalization on language tasks.

## 1 INTRODUCTION

Modularity is an important organization principle in both artificial (Ballard, 1987; Baldwin & Clark, 2000) and biological (von Dassow & Munro, 1999; Lorenz et al., 2011; Clune et al., 2013) systems. It provides a natural way of achieving compositionality, which appears essential for systematic generalization, one of the areas where typical artificial neural networks (NNs) do not yet perform well (Fodor et al., 1988; Marcus, 1998; Lake & Baroni, 2018; Hupkes et al., 2020).

Recently, NNs with explicitly designed modules have demonstrated superior generalization capabilities (Clune et al., 2013; Andreas et al., 2016; Kirsch et al., 2018; Chang et al., 2019; Bahdanau et al., 2019; Goyal et al., 2021b), which support this intuition. An implicit assumption behind such models is that NNs without hand-designed modularity do not learn to become modular by themselves. In contrast, it was recently shown that certain types of modular structures do emerge in standard NNs (Watanabe, 2019; Filan et al., 2020). However, due to defining modules in terms of activation statistics or clustering connectivity, it remains unclear whether these correspond to a *functional* decomposition.

This paper contributes new insights into the generalization capabilities of popular neural networks by investigating whether modules implementing specific functionality emerge and to what extent they enable compositionality. This calls for a *functional* definition of modules, which has not previously been studied in prior work. In particular, we consider functional modules given by subsets of weights (i.e. subnetworks) responsible for performing a specific 'target functionality', such as solving a subtask of the original task. By associating modules with performing a specific function they become easier to interpret. Moreover, depending on the chosen target functionality, modules at multiple different levels of granularity can be considered.

To unveil whether a NN has learned to acquire functional modules we propose a novel analysis tool that works on pre-trained NNs. Given an auxiliary task corresponding to a particular target function of interest (e.g., train only on a specific subset of the samples from the original dataset), we train probabilistic, binary, but differentiable masks for all weights (while the NN's weights remain frozen). The result is a binary mask exhibiting the module necessary to perform the target function. Our approach is simple yet general, which readily enables us to analyze several popular NN architectures on a variety of tasks in this way, including recurrent NNs (RNNs), Transformers (Vaswani et al., 2017), feedforward NNs (FNNs) and convolutional NNs (CNNs).

To investigate whether the discovered functional modules are part of a compositional solution, we analyze whether the NN has the following two desirable properties: ($\mathbf{P_{specialize}}$) *it uses different modules for very different functions*, and ($\mathbf{P_{reuse}}$) *it uses the same module for identical functions* that may have to be performed multiple times[1]. Here we treat $P_{specialize}$ and $P_{reuse}$ as continuous quantities, which lets us focus on the degree to which functional modularity emerges. Further, since for many tasks it is unclear what precise amount of sharing is desirable, we will measure $P_{specialize}$ and $P_{reuse}$ by considering the *change* in performance as a result of applying different masks corresponding to a target function. This yields an easy to interpret metric that does not assume precise knowledge about the desired level of weight sharing. We experimentally show that many typical NNs exhibit $P_{specialize}$ but not $P_{reuse}$. By additionally analyzing the capacity for transfer learning, we provide further insight into this issue. We offer a possible explanation: while simple data routing between modules in standard NNs is often highly desirable, it is hard to learn since the weights must also implement the data transformation. Indeed, our findings suggest that standard NNs have no bias towards separating these conceptually different goals of data transformation and information routing.

We also demonstrate how the functional modules discovered by typical NNs do not tend to encourage compositional solutions. For example, we analyze encoder-decoder LSTMs (Hochreiter & Schmidhuber, 1997) and Transformers (Vaswani et al., 2017) on the SCAN dataset (Lake & Baroni, 2018) designed to test systematic generalization based on textual commands. We show that combination-specific weights are learned to deal with certain command combinations, even when they are governed by the same rules as the other combinations. The existence of such weights indicates that the learned solution is non-compositional and fails at performing the more symbolic manipulation required for systematic generalization on SCAN. To demonstrate that this issue is present even in more real-world scenarios, we highlight identical behavior on the challenging Mathematics Dataset (Saxton et al., 2019).

Finally, we study whether functional modules emerge in CNNs trained for image classification, which are thought to rely heavily on shared features. Surprisingly, we can identify subsets of weights solely responsible for single classes: when removing these weights the performance on its class drops significantly. By analyzing the resulting confusion matrices, we identify classes relying on similar features.

## 2 DISCOVERING MODULES VIA WEIGHT-LEVEL INTROSPECTION

To investigate whether functional modules emerge in neural networks one must perform a weight-level analysis. This precludes the use of existing methods, which discover modular structure in NNs based on clustering individual *units* according to their similarity (Watanabe, 2019; Filan et al., 2020) and that may not always be enough to draw meaningful conclusions. Units can be shared even when their weights, which perform the actual computation, are not. Indeed, units can be viewed as mere "wires" for transmitting information. Consider for example a gated RNN, such as an LSTM, where gates can be controlled either by the inputs or the state, yet make use of different weights to project to the same gating units. To overcome this limitation, we propose a novel method to inspect pre-trained NNs at the level of individual weights. It works as follows. First, we formulate a target task corresponding to the specific function for which we want to investigate if a module has been learned. For example, this can be a subset of the original problem (i.e. a subtask), or based on a particular dataset split, e.g. to test generalization. Next, we train a weight mask on this target task while keeping the weights themselves frozen. The resulting mask then reveals the module (subnetwork) responsible for the target task.

To train the mask, we treat all $N$ weights seperately of each other. Let $i \in [1, N]$ to denote the weight index. The mask's probabilities are represented as learned logits $l_i \in \mathbb{R}$, which are initialized to keep the weights with high probability (0.9). If one were to apply continuous masks to the weights it would be possible to scale them arbitrarily, thereby potentially modifying the function the network performs. To prevent this, we binarize masks, which only provides for keeping or removing individual weights. The binarization is achieved using a Gumbel-Sigmoid with a straight-through estimator, which we derive from the Gumbel-Softmax (Jang et al., 2017; Maddison et al., 2017) in Appendix A.1. A

---

[1]We emphasize the distinction between the ability to *reuse* modules and the ability to compose them: a compositional solution may fail to reuse a module to implement the same behavior multiple times. Similarly, weights can be reused without them being composed to yield a compositional solution. Further, we consider *specialization* of modules a special case of *modularization* where modules are specialized to implement a particular functionality that is semantically meaningful.

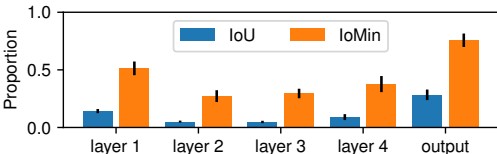 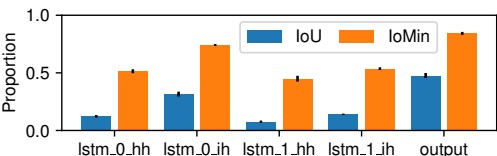

Figure 1: Proportion of shared weights per layer on addition/multiplication. Left: FNN, right: LSTM.

sample $s_i \in [0,1]$ from the mask can be drawn as follows:

$$s_i = \sigma\left(\left(l_i - \log\left(\log U_1 / \log U_2\right)\right) / \tau\right) \ \text{ with } \ U_1, U_2 \sim U(0,1), \tag{1}$$

where $\tau \in (0,\infty)$ is the temperature and $\sigma(x) = \frac{1}{1+e^{-x}}$ is the sigmoid function. Next, we can use a straight-through estimator (Hinton, 2012; Bengio et al., 2013) to obtain a binarized sample $b_i \in \{0,1\}$:

$$b_i = \left[\mathbb{1}_{s_i > 0.5} - s_i\right]_{\text{stop}} + s_i, \tag{2}$$

where $\mathbb{1}_x$ is the indicator function and $[\cdot]_{\text{stop}}$ is an operator for preventing backward gradient flow. In this case the $b_i$ are samples from a $\text{Bernoulli}(\sigma(l_i))$ random variable (proof in Appendix A.3). The masks are applied elementwise: $w_i' = w_i * b_i$. Training is done by applying the loss function defined by the target task and backpropagating into the logits $l_i$. Typically multiple (between 4–8) binary masks are sampled and applied to different parts of a batch to improve the quality of the estimated gradient.

The goal of the masking is to remove weights that are not necessary to perform the target function. Thus, the logits $l_i$ should be regularized, such that the probability for weight $w_i$ to be active is small unless $w_i$ is necessary for the task. This is achieved by adding a regularization term $r = \alpha \sum_i l_i$ to the loss, where $\alpha \in [0,\infty)$ is a hyper-parameter responsible for the strength of the regularization. How to best choose $\alpha$ is described in detail in Appendix C.3. At the end of the training process, deterministic binary masks $M_i \in \{0,1\}$ for weights $i$ are obtained via thresholding $M_i = \mathbb{1}_{\sigma(l_i)>0.5}$[2]. Applying the full mask $M$ then uncovers the module responsible for the target task. A preliminary study confirmed that the mask training process is stable and thereby suitable for inspection (Appendix B.1).

In the following sections we will analyze several standard NNs using this technique of mask-training[34]. Throughout our experiments we will avoid drawing conclusions based on the measured amount of sharing alone as much as possible, since it is unclear what degree of sharing can be expected or is desirable. Rather, we will analyze the performance drop caused by removing weights corresponding to certain functionality, which offers a more consistent and easier to interpret metric[5]. For example, to show that a module *is* responsible for a particular subtask ($A$) but not for another ($B$), we train masks on $A$ and test on both. A performance drop is expected on task $B$ only. In contrast, to show that this module *is exclusively* needed for a particular subtask, we can invert the masks and test on both tasks. The inverted masks are expected to perform well on the complementary task, but not on the original one. However, we note that this mask inversion method is limited to analyzing entirely disjoint weights.

We analyze weight sharing *between* two tasks using two different metrics: one is *Intersection over Union (IoU)*, which measures how much the weights used for solving the tasks overlap. We call the other *Intersection over Minimum (IoMin)*, which measures the number of overlapping weights (intersection) divided by the minimum of the total number of weights used for each task. In that sense IoMin is a measure of "subsetness". Intuitively, if no weights are shared, both IoU and IoMin are zero. If all weights are shared, both IoU and IoMin are one. However, when the weights needed for one task are a strict subset of the other, then IoMin is one, while IoU < 1.

---

[2]In general we find that $l_i$ concentrates at either 0 or 1 and so thresholding is safe (see also Fig. 9).

[3]A complete overview of all experimental details is available in Appendix C. Mean and standard deviations shown in the figures are calculated over 10 runs unless otherwise noted.

[4]Code for all experiments is available at https://github.com/RobertCsordas/modules.

[5]Exceptions only include cases where the observed amount of weight sharing can be clearly interpreted. However, even in these cases, our analysis will focus on general trends rather than the precise amounts observed.

## 3 Analyzing Fundamental Properties of Modules

Let us consider $P_{specialize}$ and $P_{reuse}$ (defined in Sec. 1) in more detail, as they reflect the advantages of modular compositional design. According to our notion of functional modularity, an NN is not modular without $P_{specialize}$. Moreover, disjoint modules prevent catastrophic interference (McCloskey & Cohen, 1989; Rosenbaum et al., 2019), since changing the weights responsible for a specific function does not affect the others. $P_{reuse}$ also has multiple advantages. It increases data efficiency by processing all relevant data using the same module, which thus receives additional training when a module can be reused. It also helps with generalization. For example, consider processing the expressions $a * b$ and $(c + d) * e$ where $a$, $b$, $c$ and $d$ are sampled from the same range. By reusing the multiplier it will be able to perform $a * b$ on wider range of inputs then it would otherwise be trained for.

In this section we conduct several experiments using synthetic datasets to test whether NNs have a natural inductive bias supporting $P_{specialize}$ and $P_{reuse}$. These experiments are designed to be as simple as possible to isolate the property of interest. Let's assume the network consists of compositional modules. The input of such modules can come from multiple sources within the network. Similarly, their output could be connected to different parts of the network. For example, in the previous arithmetic expression, the first operand of the multiplier can come directly from the input or the output of the adder. The same holds for the outputs. Therefore we consider cases where the inputs and outputs are shared between the modules of the ideal solution (*shared I/O*) and where they are separated (*separate I/O*).

We construct two different datasets for analyzing $P_{specialize}$ and $P_{reuse}$. For $P_{specialize}$, we use shared I/O and two different target functions (addition/multiplication task in section 3.1). The shared I/O biases the network towards weight sharing by default. Thus we use this dataset to test whether there is a bias for specializing different computations (functions) to separate weights. In contrast, to test for $P_{reuse}$, we construct a dataset where the same function should be performed twice, but using separate I/O (double addition task in section 3.2). Since separate I/O biases the network to not share weights at initialization time, we will make use of this dataset to test whether NNs exhibit a bias for reusing computation. Here reusing weights is expected, since information routing is assumed to be easier to learn than the actual function (addition). We emphasize that this initial bias due to different choices for I/O arises naturally in any network composing multiple different internal modules to arrive at a solution.

The conclusions are surprising: typical NNs tend to satisfy $P_{specialize}$ but not $P_{reuse}$. Our experiments suggest that weight sharing across tasks is mostly driven by shared I/O rather than task similarity, which results in redundancies and a lack of data efficiency.

### 3.1 Addition/Multiplication Experiments

The addition/multiplication dataset is designed to test $P_{specialize}$. The task is to add or multiply numbers (modulo 100). The input and output units are the same for both operations. An additional one-hot input specifies the operation. The numbers are two-digit and encoded as two 10-way one-hot vectors, each representing a digit. Thus, the total input is 42-dimensional, and the output is 20.

First, we train the network to perform this task without any masking. Once the performance is nearly perfect, we freeze its weights. We perform two stages of mask training: first, we train a mask on addition (multiplication examples excluded), then we repeat this procedure for multiplication.

We analyze FNN and LSTM on this task. For LSTM, we present the full input for a fixed number of timesteps. The result is the output at the final step. No loss is applied at intermediate steps. Regardless of the architecture, we found the same general tendencies: There is more sharing in the input and output layers and less in the hidden layers (Fig. 1). We also found that the multiplication uses 3.8 times more weights than the addition (Fig. 12), which partially explains the low IoU in this case. We conclude that there does appear to be some bias towards specializing weights according to different functions. On the other hand, the separation might still be inadequate to prevent interference and catastrophic forgetting. Increased sharing in I/O layers could be due to a switching/routing procedure used to select which operation to perform.

We further analyzed how performance breaks down on the task for which the mask was *not* trained on. Here the behavior of the FNN and the RNN differ. The FNN tends to ignore the function description and performs the operation for which the mask was trained, while the LSTM tends to produce invalid outputs, suggesting that it learned a solution where the two operations are more intertwined (Fig. 13).

|  |  | Full | Pair 1 | ¬Pair 1 | Pair 2 | ¬Pair 2 |
|---|---|---|---|---|---|---|
| FNN | Pair 1 | $100 \pm 0.0$ | $100 \pm 0.0$ | $20 \pm 12.7$ | $1 \pm 0.1$ | $92 \pm 10.5$ |
|  | Pair 2 | $100 \pm 0.0$ | $1 \pm 0.1$ | $94 \pm 6.7$ | $100 \pm 0.0$ | $21 \pm 11.0$ |
| LSTM | Pair 1 | $100 \pm 0.0$ | $100 \pm 0.0$ | $2 \pm 0.5$ | $1 \pm 0.1$ | $99 \pm 3.0$ |
|  | Pair 2 | $100 \pm 0.0$ | $1 \pm 0.1$ | $100 \pm 0.2$ | $100 \pm 0.0$ | $2 \pm 0.3$ |
| LSTM (forced) | Pair 1 | $100 \pm 0.0$ | $100 \pm 0.0$ | $4 \pm 0.8$ | $1 \pm 0.1$ | $99 \pm 0.7$ |
|  | Pair 2 | $100 \pm 0.1$ | $1 \pm 0.1$ | $96 \pm 4.1$ | $100 \pm 0.0$ | $3 \pm 0.6$ |

Table 1: Double-addition task: accuracy [%] of LSTMs and FNN on the two pairs. In case of LSTM (forced) only one input is presented at a time (to prevent interference). The header shows on which pair the mask was trained on. ¬ denotes an inverted mask.

## 3.2 Double-Addition Experiments

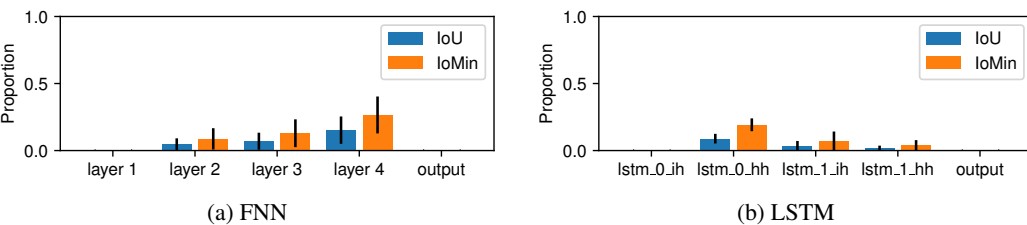

(a) FNN          (b) LSTM

Figure 2: Double addition task: proportion of weights shared per operation in case of (a) feedforward network, (b) LSTM, both inputs presented together. The first and last layers have no shared weights.

The double-addition experiment is designed to test property $P_{reuse}$. The task is to perform modulo 100 addition twice using separate I/O (different units) for each of the two instances. Using inputs $a$, $b$, $c$ and $d$, the network should output $a + b$ and $c + d$. This simulates the realistic scenario of having different data sources within a network when composing modules dynamically, without considering the additional problem of finding the right composition. Since the operation is the same and the operands' data distributions are exact matches, this simple setup encourages sharing. The encoding is the same as in section 3.1, resulting in 80 input and 40 output units.

We first train the network until convergence on the full task, then freeze its weights. We train a mask on $a + b$, followed by $c + d$. We analyze both FNN and LSTM architectures. FNN needs special care to avoid activation interference. When both operations have to be performed simultaneously, sharing is impossible. Thus, for the FNN, we perform two forward passes. In each pass, we feed only one pair of numbers to the network (either $a$, $b$ or $c$, $d$), while zeroing out the other. With LSTM, we investigate two different settings. In the first, both pairs are presented together for a fixed number of steps, and the result is the final output. Hence, the LSTM is allowed to schedule the execution of the operations freely. In the second setting, called LSTM (forced), we remove any incentive for solving the pairs simultaneously by feeding a single pair for multiple steps with the other zeroed out, and then read its output. This procedure is then repeated for the second pair without resetting the state.

All experiments' results are consistent: weight sharing is low (Fig. 2). To assert the modules' independence, we invert masks trained on pair 1, removing all weights needed for pair 1. We test the resulting network on pair 2, where the performance decreases only slightly, suggesting that they are independent (Tab. 1, further analysis is provided in Appendix C.5.1). No difference between the two LSTM variants was observed.

These observations show that $P_{reuse}$ is violated even in this simple case. Realistic scenarios tend to be more complex as the data distribution for different operation instances might be different (with overlaps), providing even fewer incentive to share. Furthermore, comparing the results to those of section 3.1, it is apparent that sharing depends more on the location of the inputs/outputs than on the similarity of the performed operations. This behavior is undesired and calls for further research.

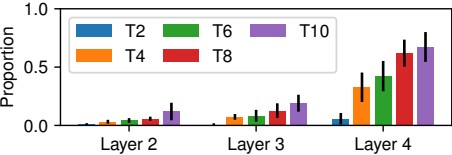

Figure 3: Proportion of the weights of a task shared with any of the previous tasks. Every second task on permuted MNIST. Each task corresponds to a permutation. The last layer has the lowest capacity, filling up first, forcing subsequent runs to share weights.

### 3.3 Transfer Learning Experiments

Let us now consider a more complex setting to assess the degree to which property $P_{reuse}$ is violated (see sec. 3.2). Here, we will measure the amount of possible transfer in a continual learning setup using the popular permuted MNIST benchmark (Kirkpatrick et al., 2017; Golkar et al., 2019; Kolouri et al., 2019). A sequence of tasks is created by applying different permutations to MNIST images (LeCun et al., 2010). Spatially close pixels may no longer be observed in nearby locations in this case, which leads us to train a FNN (as opposed to a CNN) sequentially on all permutations (tasks).

Continual learning is closely related to *transfer learning*, which is additionally concerned with transferring knowledge between tasks to improve learning and use fewer parameters. Typical approaches revolve around freezing used weights via masking when a new task is added (Fernando et al., 2017; Mallya & Lazebnik, 2018; Golkar et al., 2019). We adjust our method accordingly: We train on a single task and freeze the occupied weights. In particular, to be able to bias the network towards weight sharing, we train masks and weights simultaneously in this case. The free weights are then reinitialized and a new mask is allocated for the next task to obtain a mask for each permutation.

Note that since each task differs only by the input's permutation, it suffices to re-train a new 'first layer' to undo the permutation so that later layers can be reused. Indeed, since a significant portion of the weights is in the hidden layers, knowledge transfer between the permutations is possible and expected to be beneficial. However, re-learning the first layer may not always possible in practice since the required weights could already have been occupied to address a previous permutation. To ensure that this does not happen, we always reset the first layer and do not freeze any of its elements. Notice how, while this departs from the standard transfer learning setting, it still provides us with an *upper bound* on the amount of transfer that is possible when no such conflict occurs. Even with these modifications, we observed only little weight sharing when sufficient free space was available (Fig. 3). Only once all the capacity is saturated, weights become shared. This effect is especially apparent for the output layer.

We additionally conducted an experiment in which we explicitly bias the network towards sharing. We initialized elements of new masks corresponding to occupied weights by a significantly higher probability ($P \approx 0.88$) compared to the unused ones ($P \approx 0.27$). Intuitively, this encourages reusing the old, frozen network and adds new weights with low probability. This was able to force the network to significantly share in later layers (see Fig. 14). However, we emphasize that knowledge about which weights have to be reused between which samples is usually not available and explicitly biasing the network in this way is therefore generally not possible.

Together these observations re-affirm that $P_{reuse}$ does not emerge naturally and that the same functionality is re-learned. This is both redundant and potentially harmful as we investigate in Sec. 4.

### 3.4 A Potential Explanation for Lack of Weight Sharing

Let us consider a possible explanation for the lack of weight-sharing observed in sections 3.2 and 3.3, which is that data routing is difficult in standard NNs. Indeed, inputs and outputs must be correctly routed to different sources/targets to reuse modules in different compositions. In routing networks (Kirsch et al., 2018; Rosenbaum et al., 2019; Chang et al., 2019), this is achieved through hand-designed mechanisms. Without those, routing can only occur through the weights of the NN. However, such a 'routing transformation' would change the data representation alongside the routing unless the weights have a special structure that we empirically find is hard to learn. Indeed, our experiments suggest that NNs find it hard to learn to represent data similarly along different information routes that can in principle be processed by a single module. We argue that this is an important issue and that additional research on suitable inductive biases is needed to address this. Further discussion on the potential role of attention to mitigate this is provided in Appendix B.4.

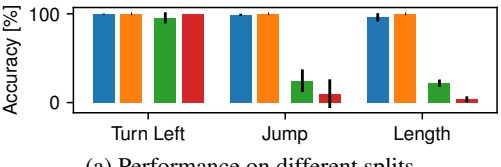 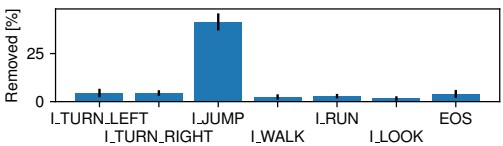

(a) Performance on different splits    (b) Weights removed from last layer on "Add jump"

Figure 4: Results of experiments on SCAN. (a) Test accuracy on split shown on $x$-axis with masks trained on the full problem (blue, orange) and with masks trained on split shown on $x$-axis (green, red). LSTM: blue, green, Transformer: orange, red (b) Percentage of weights removed per token from the output layer of the LSTM decoder trained on the "Add jump" split.

# 4  ANALYZING SYSTEMATIC GENERALIZATION ON ALGORITHMIC TASKS

Let us now consider the known issue of systematic generalization in light of our previous observations. First, what is meant by systematic generalization? Once an NN has learned to perform certain operations and some of their combinations, it should perform well on unseen combinations governed by the same algorithmic rules. Hence, it requires $P_{reuse}$ to hold. The failure of typical NNs to generalize systematically is one of the central issues of current-day NN research. The SCAN dataset (Lake & Baroni, 2018) is designed for analyzing the degree to which NNs can generalize systematically. It consists of compositional commands (e.g. "jump twice"), to be translated into primitive output moves (e.g. "JUMP JUMP"). The "simple" data split is IID, the "length" split has shorter training samples than test samples, and the "add primitive" splits have a particular command presented in the training set but no compositions of this command with others (as in the test set).

It was previously shown that typical NNs generalize poorly on data splits systematically different from the train set (Lake & Baroni, 2018; Saxton et al., 2019). The root of the problem is unclear, however. In fact, there might be two explanations: (a) The NN might have learned the correct algorithm for solving the problem, but failed to pick up on certain symmetries between concepts due to scarce evidence in the train set. For example, in the "add primitive" split, the NN might be unable to form an analogy between the additional primitive and the well-performing ones. This can also be understood as a representation problem: in this case the NN has failed to represent the new primitive in a way that lets them be used for problem solving in a similar manner following the acquired solution. However, the NN is not pressured to improve since the learned solution suffices for solving the training set. (b) Alternatively, the NN might not have learned the correct algorithm to solve the problem. For example, it may have learned to recognize patterns determining when an output token should be produced in place of reusable rules. In this case, new weights are required to solve new problems of the same kind, since they correspond to different patterns. Only in this case, we argue, has the NN failed to leverage the problem's compositional nature. Note that (a) requires $P_{reuse}$ to hold such that weights responsible for performing each individual operation are *shared* between different samples, while (b) does not.

We have tested two networks: the baseline 2 layer LSTM encoder-decoder model from Lake & Baroni (2018) and a Transformer (Vaswani et al. (2017) (see Appendix C.7.1). We pretrain the model on the IID data, which *ensures that the learned weights are capable of solving the full problem* and that a potential absence of sufficient evidence for learning about the correct symmetries between concepts is not an issue. Hence, this rules out explanation (a) being the only issue. For each split, we train a mask on its train set and measure the discovered subnetwork's performance on the corresponding systematically different test set. This process removes the weights that are not required to solve the train set for a given split. However, all splits' train sets contain sufficient information about the *full* set of rules required to perform well on *any* split. Hence, should the masking process remove any important weights, then we argue that the solution is likely pattern-recognition like rather than based on reusable rules, providing evidence for explanation (b). Indeed, our experimental results demonstrate precisely this behavior as can be seen from the large generalization gap in Fig. 4a. Note that while this gap is consistent with the findings of Lake & Baroni (2018), we are additionally able to provide evidence that the learned algorithm is likely inherently non-compositional, i.e. by eliminating explanation (a) being the only issue as a possibility.

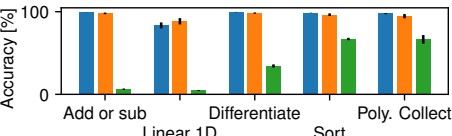

Figure 5: Accuracy on the "hard" test set of different tasks of the Mathematics Dataset: model without masks, masks trained on IID data and masks trained on "easy" set. A performance drop can be observed, because of the sample-specific weights. 5 seeds/task.

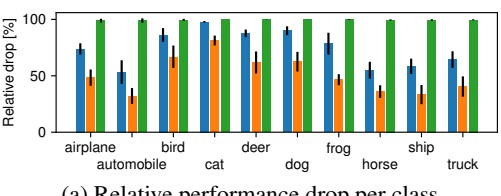

(a) Relative performance drop per class

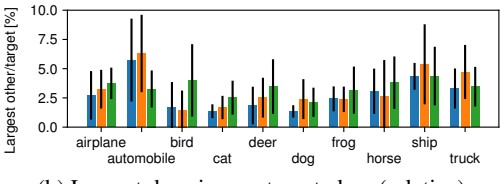

(b) Largest drop in non-target class (relative)

Figure 6: (a) Relative drop in performance for simple CNN, simple CNN without dropout and ResNet-110. (b) Largest performance drop in a non-target class relative to the drop in the target class.

To assess if the same behavior can be observed in a more complex real-world setting, we conduct a similar experiment on the challenging Mathematics Dataset (Saxton et al., 2019). Here, we generated difficulty-based splits for tasks like differentiation, solving linear equations, sorting, etc. (further details in Appendix C.7.2). In Fig. 5 a consistent performance drop can be observed when applying the inferred subnetwork on the "hard" split using a mask trained on the "easy" split. This demonstrates that samples in the "hard" split depend on exclusive weights, despite those being governed by the same underlying rules, which is consistent with results on SCAN.

The weight level analysis provided by our method enables us to gain further insight. We inspect the LSTM decoder's weights on the "add jump" split of SCAN and note that the most apparent difference is in the output layer. Almost half of the weights corresponding to "I_JUMP" are removed (Fig. 4b), suggesting that the network learned to detect patterns of cases when "I_JUMP" should be the output, and the last layer puzzles them together. In contrast, we hypothesize that the generalizing algorithm for solving such problems necessitates proper variable manipulation (Garnelo & Shanahan, 2019).

## 5 ANALYZING CONVOLUTIONAL NEURAL NETWORKS

As a final case-study, we consider whether we are also able to observe a lack of weight-sharing in CNNs. By conducting a weight-level analysis using our tool, we are able to highlight sets of non-shared weights solely responsible for individual classes. We consider multiple CNN architectures trained on CIFAR10 (Krizhevsky et al., 2009): a simple CNN with dropout (an ablation is provided in Appendix C.8.2) and a ResNet-110 (He et al. (2016). Full details are available in Appendix C.8). We proceed as follows. First, we train a 'control mask' on the full dataset to highlight all used weights. Next, we train a mask with a single class removed so that the weights solely responsible for this class will be absent from the resulting mask. Here we avoid removing all weights responsible for the this class from the output layer (leaving no connection to the corresponding output unit) by fixing its mask to one trained on the full dataset. This corresponds to inspecting the feature detector layers as opposed to the classifier. We repeat this process for all classes to obtain a total of 11 masks.

We compute the confusion matrix on the full validation set at the end of each stage. Then we calculate the difference between the confusion matrices with and without the removed class, which unveils how the removal changes the classification. Interestingly, the performance of the target class drops significantly (Fig. 6a), only a small drop in performance (possibly due noise when mask sampling) is observed for non-target classes (Fig. 6b). This indicates a large dependence on class-exclusive, non-shared weights in the feature detectors. These findings, which assume that the network has sufficient capacity relative to dataset size, are in line with those observed in sections 3–4. Analyzing the difference in misclassification rates yields further insights: as the true positive rate drops, certain other classes are predicted instead that appear to rely on similar shared features. For example, removing "airplane" causes images to be classified as "birds" and "ships" instead, which have a blue background in common. Additional insights are reported for other classes in Fig. 18 in Appendix C.8.

## 6  RELATED WORK

There have been few other attempts at analyzing emerging modularity in NNs. Filan et al. (2020) identifies groups of neurons with strong internal and weak external connectivity via clustering, while others group neurons based on their connectivity pattern (Watanabe et al., 2018) or cluster them hierarchically based on activation statistics (Watanabe, 2019). However, as we have argued, without considering the contribution of individual weights it is not always possible to reason about *functional* modularity. Davis et al. (2020) considers an alternative approach based on mutual information to detects salient pathways in NNs that could in principle allow for this. However, the discovered pathways are not grounded with respect to particular functionality, nor is it analyzed whether they support compositionality. Bengio et al. (2015) formulate adaptive mask learning as a reinforcement learning problem, with the main goal of accelerating inference via conditional execution. However, the masking is unit level and trained together with network weights. Similarly, functional modularity is not considered.

It has repeatedly been argued that NNs lack compositionality due to their failure at systematic generalization (Lake & Baroni, 2018; Bahdanau et al., 2019; Barrett et al., 2018; Hupkes et al., 2020; Hill et al., 2020). Typically this analysis proceeds by evaluating NNs on a dataset that exhibits some systematic difference from the training data, yet can be solved through clever recombination of known concepts based on inferred rules. Here we were additionally able to show that the learned solution incorporates combination-specific weights even on the training set, suggesting that this issue runs deeper than simply being unable to learn about symmetries between well-known and novel concepts. Andreas (2019) introduced a direct measure of compositionality by reconstructing NN representations from a set of primitive representations and a learned composition function. However this method is concerned about the data representation and not about the modularity of the performed computation.

Finally, we note how many transfer and continual learning methods make use of weight freezing via masking to prevent catastrophic forgetting (Fernando et al., 2017; Mallya & Lazebnik, 2018; Golkar et al., 2019; Yang et al., 2020). Determining the importance of individual weights has been studied in network pruning (LeCun et al., 1990; Hassibi & Stork, 1993; Li et al., 2017; Frankle & Carbin, 2019; Gaier & Ha, 2019) and feature attribution (Simonyan et al., 2013; Springenberg et al., 2015; Sundararajan et al., 2017; Shrikumar et al., 2017) often using weight and/or gradients magnitudes. Differentiable binary weight masks have also been explored in the multi-task setting (Mallya et al., 2018), albeit deterministically in contrast to the Gumbel-Sigmoid used here. It should also be mentioned how many explicitly modular architectures have been proposed to improve generalization (e.g. Clune et al. (2013); Andreas et al. (2016); Kirsch et al. (2018); Chang et al. (2019); Goyal et al. (2021b)) and data efficiency (Purushwalkam et al., 2019). Rather than engineering an explicitly modular solution, our goal is to let this emerge naturally. We believe that our current findings help take a step in that direction.

## 7  CONCLUSION

Our new method for inspecting modularity in neural networks is the first to identify modules by their functionality. It is a powerful tool for analyzing how the NNs share or separate weights based on the performed computation. By analyzing diverse sets of neural networks (FNNs, CNNs, RNNs, Transformers), we could draw significant novel conclusions: in typical current NNs, weight sharing between modules does not reflect task similarity (as desired) but can mostly be explained by rather trivial shared I/O interfaces of solution-implementing modules. The lack of weight sharing between multiple uses of the same function makes the learning data inefficient since it has to be re-learned repeatedly. Moreover, NNs trained on algorithmic tasks appear to fail to learn general, modular, compositional algorithms. Rather, we have shown that they require specific subset weights to solve a particular combination of the input tokens, even when the same rules govern both the solution as the other samples. Our discoveries call for future research: function dependent weight sharing in the neural networks should vastly improve data efficiency, and encouraging algorithmic solutions should improve generalization.

### ACKNOWLEDGMENTS

We wish to thank Aleksandar Stanić, Francesco Faccio, Kazuki Irie & Louis Kirsch for their constructive feedback. This research was supported by a European Research Council Advanced grant (no: 742870), two Swiss National Science Foundation grants (no: 200021_165675/1, 200021_192356) and hardware donations from NVIDIA & IBM. We thank Weights & Biases for a free academic license.

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

## A DERIVATIONS

### A.1 FROM GUMBEL-SOFTMAX TO GUMBEL-SIGMOID

In what follows, we use the notation by Jang et al. (2017): $k$ is the number of categories, class probabilities are $\pi_i$, $y \in \mathbb{R}^k$ is a sample vector from the Gumbel-Softmax distribution (also called Concrete distribution by Maddison et al. (2017)). Individual components of $y$ are denoted by $y_i$, $i \in [1, k]$. We will refer to $l_i = \log \pi_i$ as logits. We show how to sample from the Gumbel-Sigmoid distribution, the special case of $k = 2$, $l_2 = 0$ of the Gumbel-Softmax distribution.

First, we show that the sigmoid is equivalent to a first element $y_1 \in \mathbb{R}$ of the output vector of the two element softmax with $l_1 = x$, $x \in \mathbb{R}$ and $l_2 = 0$:

$$\sigma(x) = \frac{1}{1 + e^{-x}} = \frac{e^x}{e^x + 1} = \frac{e^x}{e^x + e^0} = \frac{e^{l_1}}{e^{l_1} + e^{l_2}} = y_1. \tag{3}$$

According to Jang et al. (2017), the sample vector $y \in \mathbb{R}^k$ from the Gumbel-Softmax distribution can be drawn as follows:

$$y_i = \frac{e^{\frac{1}{\tau}(l_i + g_i)}}{\sum_{j=1}^{k} e^{\frac{1}{\tau}(l_j + g_j)}}, \tag{4}$$

where $g_i \sim \text{Gumbel}(0, 1)$ are independent samples from the Gumbel distribution. We are interested in the special case of a sigmoid, which we showed to be equivalent to the $y_1$ in $k = 2$, $l_2 = 0$ case:

$$y_1 = \frac{e^{\frac{1}{\tau}(l_1 + g_1)}}{e^{\frac{1}{\tau}(l_1 + g_1)} + e^{\frac{1}{\tau} g_2}}. \tag{5}$$

This can be rearranged as:

$$y_1 = \frac{1}{1 + e^{-\frac{1}{\tau}(l_1 + g_1 - g_2)}} = \sigma\left(\frac{1}{\tau}(l_1 + g_1 - g_2)\right). \tag{6}$$

Writing out the inverse transformation sampling formula for $g_i \sim \text{Gumbel}(0, 1)$; $g_i = -\log(-\log U_i)$, were $U_i \sim \text{U}(0, 1)$ are independent samples from the uniform distribution, we get:

$$
\begin{aligned}
y_1 &= \sigma\left(\frac{1}{\tau}(l_1 - \log(-\log U_1) + \log(-\log U_2))\right) \\
&= \sigma\left(\frac{1}{\tau}\left(l_1 - \log \frac{\log U_1}{\log U_2}\right)\right).
\end{aligned}
\tag{7}
$$

Finally, by renaming $s = y_1$ and $l = l_1$ (we have just a single logit), we obtain the sampling formula for Gumbel-Sigmoid:

$$s = \sigma\left(\frac{1}{\tau}\left(l - \log \frac{\log U_1}{\log U_2}\right)\right). \tag{8}$$

### A.2 STRAIGHT-THROUGH ESTIMATOR

Samples from the Gumbel-Softmax distribution can directly be converted to a sample from the categorical distribution as:

$$c_i = \mathbb{1}_{i = \arg\max_i y_i}. \tag{9}$$

Applying the straight-through estimator (Hinton, 2012; Bengio et al., 2013) can provide 'hard' samples while permitting gradient flow ($[\cdot]_{\text{stop}}$ is an operator for blocking backward gradient flow):

$$c_i = [\mathbb{1}_{i = \arg\max_i y_i} - y_i]_{\text{stop}} + y_i. \tag{10}$$

Since in the Gumbel-Sigmoid we have $k = 2$ categories and $\sum_i y_i = 1$, the $\arg\max$ can be replaced by testing whether $y_i > 0.5$:

$$c_i = [\mathbb{1}_{y_i > 0.5} - y_i]_{\text{stop}} + y_i. \tag{11}$$

Since we defined the sample $s$ to be $y_1$ (i.e. we are interested in the $i = 1$ case) the index $i$ can be omitted. The variable has a Bernoulli distribution (see Section A.3), so we use a substitution $b = c_1$ to get:

$$b = [\mathbb{1}_{s > 0.5} - s]_{\text{stop}} + s. \tag{12}$$

### A.3 THE EXPECTED VALUE OF THE SAMPLES

Let us analyze the distribution governing the samples $b$. Each sample is binary and independent since it is generated using independent samples from a uniform distribution. Since the sampling process is stationary they must therefore be Bernoulli distributed. Next, we show that their mean is $\mu = \sigma(l)$.

We are interested in:
$$\mu = P(b = 1) = P\left([\mathbb{1}_{s>0.5} - s]_{\text{stop}} + s = 1\right). \tag{13}$$

The straight-through estimator does not change the numerical value of $b$, so we can ignore it:
$$\mu = P(b = 1) = P\left(\mathbb{1}_{s>0.5} = 1\right) = P\left(s > 0.5\right), \tag{14}$$

where $s = \sigma\left(\frac{1}{\tau}\left(l - \log\frac{\log U_1}{\log U_2}\right)\right)$. Let us simplify the condition $s > 0.5$:
$$\sigma\left(\frac{1}{\tau}\left(l - \log\frac{\log U_1}{\log U_2}\right)\right) > 0.5 \tag{15}$$

$\sigma$ is monotonically increasing and $\sigma(0) = 0.5$, so:
$$\frac{1}{\tau}\left(l - \log\frac{\log U_1}{\log U_2}\right) > 0. \tag{16}$$

By multiplying both sides with $\tau > 0$ and re-ordering we obtain:
$$l > \log\frac{\log U_1}{\log U_2}. \tag{17}$$

Since $e^x$ is monotonically increasing, we can exponentiate both sides to obtain:
$$e^l > \frac{\log U_1}{\log U_2}. \tag{18}$$

The samples $U_i$ are uniform random samples from the range $U_i \in (0, 1)$. Hence, it follows that $\log U_i < 0$. Multiplying both sides by $\log U_2$, we get:
$$e^l \log U_2 < \log U_1. \tag{19}$$

Exponentiating once again leads to:
$$U_2^{e^l} < U_1. \tag{20}$$

Let us return to the original problem, which is now takes a much simpler form:
$$\mu = P(b = 1) = P(U_2^{e^l} < U_1). \tag{21}$$

Using the definition of the mean, we get:
$$\mu = \mathbb{E}_{U_1 \sim U(0,1), U_2 \sim U(0,1)}\left[\mathbb{1}_{U_2^{e^l} < U_1}\right]. \tag{22}$$

Using the definition of expectation:
$$\mu = \int_{-\infty}^{\infty} P(U_2) \int_{-\infty}^{\infty} P(U_1) \mathbb{1}_{U_2^{e^l} < U_1} dU_1 dU_2. \tag{23}$$

Since $U_i \sim U(0, 1)$ are samples form uniform distribution with range $(0, 1)$, $P(U_i) = 1$ in interval $U_i \in (0, 1)$ and 0 otherwise. This enables us to change the boundaries of the integrals:
$$\mu = \int_0^1 \int_0^1 \mathbb{1}_{U_2^{e^l} < U_1} dU_1 dU_2. \tag{24}$$

Since the value of $\mathbb{1}_{U_2^{e^l} < U_1}$ is 1 when $U_1 > U_2^{e^l}$ and 0 otherwise, we can tighten the bounds of integration and eliminate the indicator function:
$$\mu = \int_0^1 \int_{U_2^{e^l}}^1 1 dU_1 dU_2 = \int_0^1 [U_1]_{U_2^{e^l}}^1 dU_2. \tag{25}$$

$$\mu = \int_0^1 1 - U_2^{e^l} \, dU_2 = 1 - \int_0^1 U_2^{e^l} \, dU_2$$

$$= 1 - \left[ \frac{U_2^{e^l+1}}{e^l + 1} \right]_0^1 = 1 - \frac{1^{e^l+1}}{e^l + 1} + \frac{0^{e^l+1}}{e^l + 1} \tag{26}$$

$$= 1 - \frac{1}{e^l + 1} = \frac{e^l + 1}{e^l + 1} - \frac{1}{e^l + 1} = \frac{e^l}{e^l + 1} = \frac{1}{e^{-l} + 1}$$

$$= \sigma(l).$$

### A.4 CHOOSING THE TEMPERATURE

Notice that $\mu = \sigma(l)$ does not depend on the temperature $\tau$ (Appendix A.3). The binarized sample, $b$, will have the same output regardless of the value of $\tau$, thus $s$ will have the same gradients. The logit $l$, however, has a gradient scaled by $\frac{1}{\tau}$, but which can be mitigated by the normalization in the Adam optimizer. Thus, we can choose $\tau$ freely. We set $\tau = 1$ for all experiments of our paper.

## B ADDITIONAL DISCUSSION

### B.1 STABILITY OF THE MASKS

Multiple sources of randomness could affect the final masks discovered by our method. These include sampling the mask at each iteration, different data for each target task, and the order in which data is used for training. To verify that the masks discovered by our method are consistent we considered pairs of CIFAR10 classes as target tasks in combination with a simple CNN without dropout (Appendix C.8). Pairs are chosen instead of the leave-one-out scheme used in Section 5 to increase the sparsity of the masks as much as possible (potentially making them even more unstable). We trained 10 CNNs and analyzed 10 random pairs of classes for each of them. For each pair we trained two separate masks and calculate the Intersection over Union (IoU), resulting in a total of 100 data points.

We found that the mean IoU is $93.26 \pm 0.96\%$, which confirms the discovered masks' stability. Note that in case multiple redundant weight configurations are present in the network, different mask seeds will find a different subset of them, so their IoU will be less then $100\%$ even in the optimally stable case. Using dropout would encourage such cases.

#### B.1.1 POTENTIAL ERRORS INTRODUCED BY THE STRAIGHT-THROUGH ESTIMATOR

The straight-through estimator introduces approximations in the gradient calculation. Fortunately, the inaccuracies do not build up through multiple estimation steps, since the masking and straight-through estimator are applied directly to the network's parameters. Indeed, on each gradient path, there is at most a single straight-through estimator present.

### B.2 DOES MASKING CHANGE THE PERFORMED OPERATION?

The recent work of Zhou et al. (2019) demonstrated how it is possible to achieve non trivial performance by training binary masks on a neural network with frozen weights that were randomly chosen. This raises the question whether the masking process in our method changes the performed operation after the weights are frozen and could thus cause misleading observations.

To investigate this possibility, we randomly selected some of the networks and datasets used throughout the paper, and trained as usual. After both the weights and the masks are learned, we performed the following experiment: we applied masks to roughly half of the networks weights, while leaving the remainder unmasked. In one variant of this experiment, early layers near to the input are masked, while the later layers, including the output, are not. In the other variant, the opposite is true. If the network demonstrates compatibility between the masked and non-masked layers for these experiments, then this is a strong indication that it has not altered the performed operation significantly.

The outcome of these experiments are shown on Fig. 7, where we report the performance drop for transformer on SCAN dataset, FNN on addition/multiplication dataset, LSTM, big (4 layers of 2000

units) and small (4 layers of 800 units) FNNs on the double-addition experiments and the small CNN on CIFAR 10.

For almost all configurations, we observe only a low drop in performance, indicating that the operations performed by the network remain mostly the same under the masking process. The only exception we found is the big FNN on the double-addition task, when the early layers are masked. Note, however, that its performance is well above the chance level ($P = 10^{-4}$). Since this network is severely overparametrized, we speculate how this might be the reason for this observed difference. For example, it could have learned to solve the problem by combining multiple alternative pathways, all of which contribute to the output. If the masking process removes some of those pathways from the layer near the input, but leaves them in the layers near the output, the pathways are cut in half. Thus, they might produce erroneous outputs. To further analyze this we therefore also trained a smaller version of the same network, which we observe behaves similarly to all other networks, suggesting that also in this case the masking does not alter the performed computation significantly.

Finally, we note the variant where the early layers are masked appears considerably more difficult than other way around. This might be because some of the inputs of the unmasked later layers are removed by masking the early layers. Normally, if all layers are masked as well, such weights of later layers would be removed together with the ones in the early layers, thus not affecting the result.

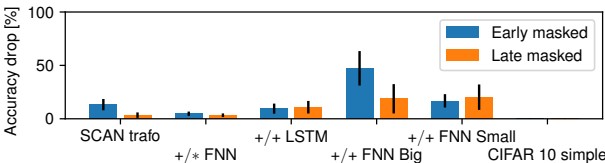

Figure 7: Accuracy drop for masks applied to half of the weights. See Sec. B.2 for details.

There may be multiple reasons for these different findings compared to Zhou et al. (2019). First, we use well trained networks instead a randomly initialized ones. Untrained network are believed to contain many random subnetworks which can useful for performing any task. However, the fully trained network has its subnetworks tuned to the task, likely decreasing the possibility of further subnetworks existing that implement radically different operations. Second, we are training the masks on a subset of the train set, which does not encourage changing the performed operations either: the highest performance can be achieved by selecting the correct subnetwork already performing the operation well. Finally, the experiments of Zhou et al. (2019) indicated that the performance of the best found network decreases with the task complexity, and performs best on MNIST. Some of the experiments considered here are significantly more complex.

### B.3 CHOOSING TARGET FUNCTIONALITY

In principle any operation that the network is able perform can be used as a target functionality. This includes partitions of the dataset, or even novel tasks if the network can generalize to them. The resulting masks will highlight which weights are responsible for performing them. For our experiments, we always chose a subset of the training set of the weights as target functionality. This ensures that no generalization is required from the network to solve the problem, and that the subset of the original weights required to solve this subproblem is highlighted. The discovered module then corresponds to functionality that the network should already have learned in the original training phase.

Interesting target functionality should be chosen such that removing the discovered set of weights or its inverse can be expected to lead to measurable performance difference on some test set. This test set should ideally be a subset of the original training set used for the weights. In this way, one does not have to directly consider the amount of sharing, and can measure (the difference in) accuracy, which we find more reliable and easy to interpret (see also Sec. 2 for further details) However, if such a choice is not available and the amount of sharing has to be analyzed directly, then we recommend drawing conclusions only when the observed difference compared to some reference score is sufficiently high. For example in the Permuted MNIST experiments, the sharing of $< 20\%$ is significantly lower than the expected $100\%$.

### B.4 IS ATTENTION THE SOLUTION?

Could a form of attention (Bahdanau et al., 2015) solve the problem discussed in Section 3.4? At least the current use of attention does not seem promising. In theory, attention-based Transformers (Vaswani et al., 2017) can reuse the same modules in parallel, but only if they are executed in the same layer. For $a*b+c*d$ the multiplier is reusable, but for $a*b*c$ it is not, since the second multiplication requires the result of the first; that is, different layers are needed. In recurrent models, such as RNNs with attention (Bahdanau et al., 2015) and Universal Transformers (Dehghani et al., 2019), attention does not permit routing between functional modules but is just used to route data to the input of a monolithic transformation block which processes the information. Emerging functional modules must in that case appear within the processing blocks. However, since attention is neither able to rewire the block's internal data flow, nor to permute elements of the attended vector, it does not help with the routing between modules emerging inside the block. Indeed, we showed empirically in Section 4 that Transformers suffer from the same generalization issues as LSTM on the SCAN dataset, and they did not generalize on the more complex Mathematics Dataset either. Attention might help though, if all the input and output interfaces of functional blocks overlap, and a single processing step executes a single function. However, as our experiments show, the separation between modules tends to be inadequate in the case of shared interfaces (Sec 3.1). Moreover, there is no control over executing a single function per time step (e.g., the whole $a*b*c$ block could be executed in a single step).

In order to help with data routing between emerging functional modules, attention has to be able to focus on arbitrary parts of the activation vectors. This would enable information exchange between such modules, but it is unclear how this could be implemented. For example, in the double addition experiment (Section 3.2), the task requires to process disjoint subsets of the input, which is not possible with the attention mechanism. In general, attention-based solutions would require to store one "concept" in a single vector so that they can separately attended to. However, what makes a good "concept" in this case is unclear, especially since different processing stages might require a different granularity – for example, sorting tuples of numbers based on the first element of the tuple requires accessing individual elements of it but also treating the tuple as a whole. For a broader discussion on dynamic information routing and the problem of variable binding in neural networks, we refer the reader to Greff et al. (2020).

### B.5 EXPLICITLY MODULAR NETWORKS

At first glance, explicitly modular networks (Clune et al., 2013; Andreas et al., 2016; Kirsch et al., 2018; Chang et al., 2019; Bahdanau et al., 2019) could provide a solution for the discovered problems. In what follows we will call hardcoded modules "blocks" to distinguish from functional modules, which we call "modules". Routing networks, in addition to the problems described by Rosenbaum et al. (2019), are restricted to exchange information between blocks as a single fixed-size activation vector. Because all the information has to be stored in this vector (such as different variables), either different parts of the vector should be responsible for different stored variables, or they have to be stored in superposition, e.g., by projecting them in a space where they are orthogonal to each other. Either way, this requires the blocks not only to perform a given operation, but also to be aware which variable they want to access. Thus, the blocks are not universal since operating on different variables encoded in the state require different modules. For example, in the double addition experiment (Section 3.2), the task requires to process disjoint subsets of the input. This is true in general: different subsets of the network state may require independent processing. Routing networks consist of simple modules stacked sequentially, which is obviously not a good fit for this type of data. Alternatively, RIMs (Mittal et al., 2020; Goyal et al., 2021b;a) attend to the data, making it possible to execute multiple modules in parallel. However, they are based on attention, which also has its limitations (Sec. B.4). These difficulties let us believe that a general inductive bias towards function-based specialization in generic neural networks would be a preferable solution compared to explicit modality and motivates this paper's topic.

## C    ADDITIONAL RESULTS AND EXPERIMENTAL DETAILS

### C.1    SANITY CHECKING THE MASK DISCOVERY PROCESS

Our method frequently discovers a resistance against weight sharing. Perhaps this could raise the question whether our method is able to discover shared weights at all. We ran additional experiments to verify this.

We used the double-addition experiments from Sec. 3.2. Specifically, we trained the network as before, but after the weight training phase, we copy the input and output weights of one pair to the part of the weight matrix corresponding to the other. This ensures that the hidden layers can not see any difference between the two pairs. We use the FNN variant since it can be used without any modification, while the LSTM would require changes to avoid state conflicts.

In Fig. 8 it can be seen how our method accurately discovers that the sharing is almost perfect in this case, which further justifies our approach. Compare this to the identical setup of Fig. 2. Note that the first and last layers are still not shared: they contain identical, but non-shared copies of the weights.

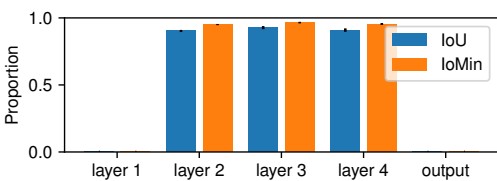

Figure 8: Double addition task with manually edited input/output weight matrices to reuse the hidden layers. Proportion of weights shared per operation in case of FFN.

### C.2    COMMON HYPERPARAMETER CHOICES

Our method is implemented in PyTorch (Paszke et al., 2019), and available at `https://github.com/RobertCsordas/modules`. Unless otherwise noted we use the Adam optimizer (Kingma & Ba, 2015), a batch size of 128, a learning rate of $10^{-3}$, and gradient clipping of 1. To improve the quality of the masks we divide a batch into four parts that each act on a different mask sample. For non-LSTM networks, we use the ReLU activation function (von der Malsburg, 1973; Nair & Hinton, 2010) for the activations of intermediate layers. The Gumbel-sigmoid always has a temperature of $\tau = 1$ (the reason for this is explained in Appendix A.4). For most of our experiments, the regularization coefficient $\alpha$ is specified as $\beta = b\alpha$, where $b$ is the batch size used for training the masks. Otherwise we will mention $\alpha$ separately. All figures in this paper, unless noted otherwise, show mean and standard deviation calculated over 10 runs with different seeds.

### C.3    CHOOSING THE REGULARIZATION HYPERPARAMETER

Choosing the regularization hyperparameter $\alpha$ is critical to obtain valid conclusions. Too low $\alpha$ might yield the false impression that no modules exist or that they share more weights than they really do. Too strong regularization may degrade performance on the target task, discarding essential weights.

Fortunately, there is a simple and consistent heuristic for choosing $\alpha$, which follows from training a mask on the full task. We increase $\alpha$ as long as the performance does not start to drop. Then, we reduce $\alpha$ slightly until the performance is adequate, e.g. $> 95\%$ of the original performance. This method is not very sensitive to the *exact* value $\alpha$ and transfers well across different network sizes (Fig. 10). We find that it is less critical but still important to tune the learning rate and the number of steps of the mask training process. We always check the chosen hyperparameters' validity by training a mask on the full, unmodified problem, where we expect to see only a slight drop in performance.

Note that underfitting NNs tend to share more weights. Indeed, in our experiments we found that choosing a sufficiently large network size is essential to avoid false conclusions about the reason for sharing. Fig. 11 shows an example how the amount of weight sharing changes as a function of network capacity.

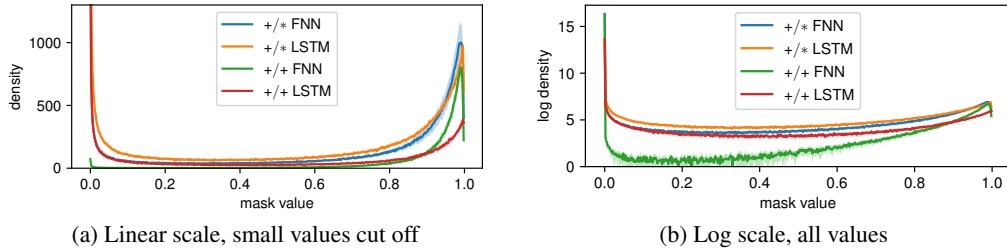

(a) Linear scale, small values cut off       (b) Log scale, all values

Figure 9: Histogram (normalized as a 500-bin PDF) of expected values of the mask elements ($\mu_i = \sigma(l_i)$) on different tasks. (a) Shown on a linear scale. Values $< 0.0002$ (bottom 10% of the first bin) are removed from the calculation because their number vastly exceeds the number of kept weights for most of the networks, making the details invisible. (b) All mask means, $\mu_i$, (without small values removed) shown on log-scale.

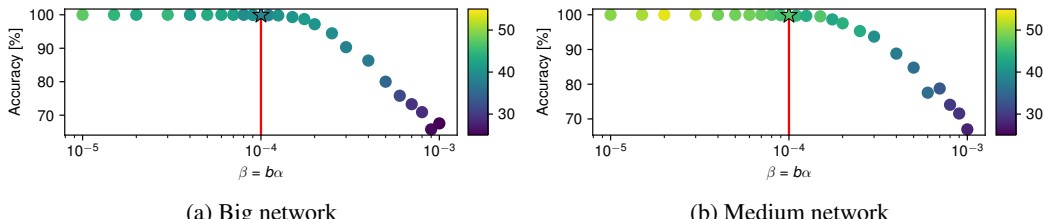

(a) Big network       (b) Medium network

Figure 10: Sensitivity analysis for hyperparameter $\beta = b\alpha$ ($b$ is the batch size) on addition/multi-plication experiments. Note the logarithmic $x$-axis. The color indicates the total amount of sharing [%]. The red line and the star indicate the value chosen for our experiments. Each point is a mean of 10 independent seeds. The network is not very sensitive to the exact choice of $\beta$. (a) Big network, with 5 layers of size 2000. (b) Medium network, with 5 layers of size 800. It can be seen that the hyperparameter transfers well between network sizes.

## C.4 Addition/Multiplication Experiments

Since preliminary experiments indicated that modulo 100 multiplication require lots of weights, we used reasonably large networks for this experiment. The FNN is 5 layers deep, each layer having 2000 units and the LSTM a hidden state size of 256 (further increase resulted in overfitting). A network was trained for 20k steps on the full task before freezing. The following mask training phase takes an additional 20k steps for each mask. Mask training uses a learning rate of $10^{-2}$ and $\beta = 10^{-4}$ for regularization. For the LSTM we use 3 time steps where the input is repeated for every step. The dataset is generated by sampling numbers and operations uniformly at random.

Fig. 11 demonstrates that even if we use a large network (the small, 3 layer networks of 400, 400, 200 can solve the task), the percentage of shared weights still changes when increasing the network size.

In Sec. 3.1 we showed that even though there is a certain level of natural separation between the modules responsible for addition and multiplication, there is still a significant proportion of shared weights. To analyze the importance of those shared weights, we tested the network with inverted masks as in Sec. 3.2. Tab. 2 shows the results. Given the high proportion of sharing, especially in the input and output layers, the results are as expected: inverted masks do not perform well, showing that the separation of the modules is limited.

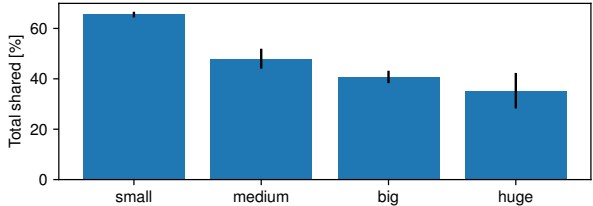

Figure 11: Addition/multiplication task: the total proportion of shared weights for the "add" operation for different network sizes. "small" means a 4 layer network with hidden sizes of 400, 400, 200, "medium" 5 layers / hidden sizes of 800, "big" 5 layers / 2000, "huge" 5 layers / 4000.

|  |  | Full | $+$ | $\neg+$ | $*$ | $\neg*$ |
|---|---|---|---|---|---|---|
| FNN | $+$ | $100 \pm 0.0$ | $100 \pm 0.0$ | $13 \pm 5.5$ | $1 \pm 0.0$ | $20 \pm 7.0$ |
|  | $*$ | $100 \pm 0.2$ | $0 \pm 0.0$ | $69 \pm 9.7$ | $100 \pm 0.0$ | $17 \pm 5.8$ |
| LSTM | $+$ | $100 \pm 0.0$ | $100 \pm 0.0$ | $2 \pm 0.6$ | $2 \pm 0.6$ | $1 \pm 0.2$ |
|  | $*$ | $100 \pm 0.1$ | $3 \pm 1.2$ | $6 \pm 0.8$ | $100 \pm 0.0$ | $2 \pm 1.2$ |

Table 2: Accuracy of addition/multiplication task on addition and multiplication with FNN and LSTM. The header shows on what the applied mask was trained on. $\neg$ denotes an inverted mask

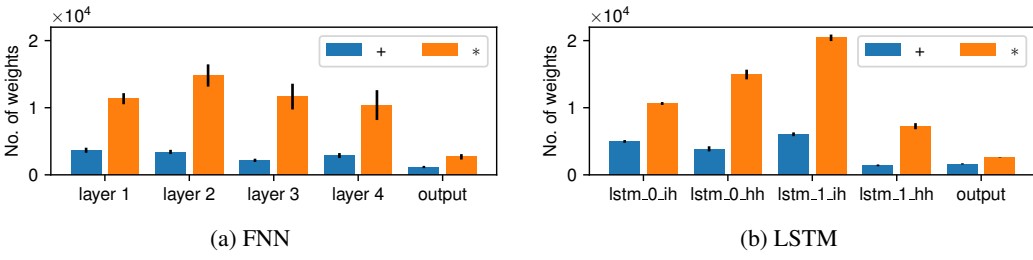

(a) FNN

(b) LSTM

Figure 12: Addition/multiplication taks: number of weights per operation for each layer in (a) feedforward network, (b) LSTM.

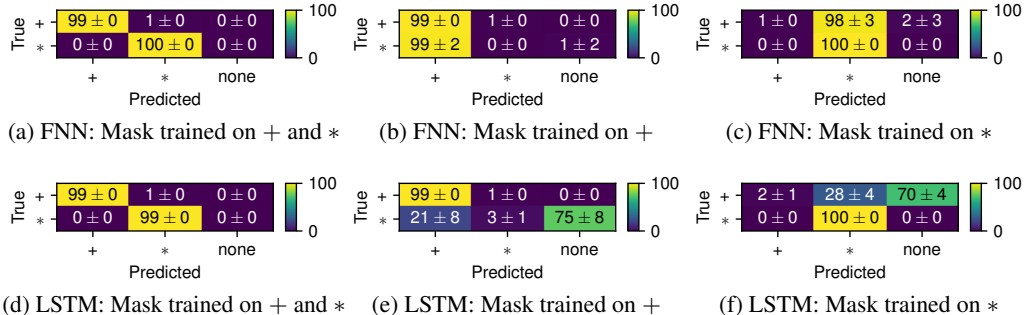

Figure 13: Analysis of FNN (a, b, c) and LSTM (d, e, f) performance degradation on the addition/ multiplication task. The $y$-axis shows the target operation. The $x$-axis shows the actual operation performed. "none" means the predicted number is neither the result of addition nor multiplication. The FNN ignores the operator specification and performs the one corresponding to the mask; in contrast, the LSTM tends to perform invalid operations.

## C.5    DOUBLE ADDITION EXPERIMENTS

The training protocol for double-addition experiments is identical to the one in Appendix C.4, except that the FNN variant uses a mask regularizer of $\beta = 4 * 10^{-4}$. The LSTM uses 6 steps in total in this case (3 steps per operation). In the full input case, both tuples are presented for all 6 steps and the output is read from the last step. In the case where one tuple is presented at a time, the first tuple is shown for the first 3 steps, resulting in an output at the $3^{rd}$ step, the second tuple is presented for the next 3 steps, resulting in an output at the $6^{th}$ step.

### C.5.1    ADDITIONAL INVERTED MASK EXPERIMENTS

Following the inverted-mask experiments done in Sec. 3.2, we investigated how well the separation holds if we consider only the hidden layers. This is achieved by using inverted masks for the hidden layers, while using the mask trained on the full task without inversion for the input and output layers. Hence, in this case the inputs and outputs contain all the connections needed for both tasks. Our findings are shown in Tab. 3 and are consistent with Tab. 1. It can be seen how the performance of the inverted mask tends to work well on the opposite task, while its performance is significantly lower on the original task (note that chance is at $P = 0.01$ for these experiments), suggesting that this effect is not due to their inputs/outputs being disjoint.

Further we experimented with leaving the input and output layers unmasked, while inverting the discovered masks for the hidden layers. Surprisingly, in this case the inverted masks perform well on both tasks (around $90\%$), even on the task for which the mask was inverted. This suggests that the network contains an ensemble of subnetworks individually capable of solving the problem with good performance. However based on the findings in Tab. 3, these subnetworks in the hidden layers appear to be mostly independent of each other: the performance is nonzero on the original task *only* if both the original weights and the ones corresponding to the inverted masks are now included. It remains unclear what causes this particular behavior in this setting, which we believe is an interesting direction for future research.

## C.6    TRANSFER LEARNING EXPERIMENTS

In the transfer learning setup, we train on 11 permutations of MNIST using the same network. Training the weights and masks together is more difficult than the usual setup. In order to improve the quality of the mask gradients we use 8 mask samples per batch instead of the standard 4. Each phase takes 30k steps. The learning rate is $10^{-2}$. The network is 4 layers deep, with hidden sizes of 800, 800, 64. We are using a mask loss of $\alpha = 10^{-5}$.

Fig. 14 demonstrates the number of shared weights per layer for a network that has its masks initialized such that it prefers to reuse the old weights. The mask logits corresponding to weights

|  |  | Full | Pair 1 | ¬Pair 1 | Pair 2 | ¬Pair 2 |
|---|---|---|---|---|---|---|
| FNN | Pair 1 | $100 \pm 0.4$ | $100 \pm 0.0$ | $7 \pm 4.0$ | $1 \pm 0.1$ | $63 \pm 15.9$ |
|  | Pair 2 | $100 \pm 0.1$ | $1 \pm 0.1$ | $62 \pm 16.9$ | $100 \pm 0.0$ | $8 \pm 5.0$ |
| LSTM | Pair 1 | $100 \pm 0.0$ | $100 \pm 0.0$ | $16 \pm 4.1$ | $1 \pm 0.1$ | $99 \pm 1.3$ |
|  | Pair 2 | $100 \pm 0.0$ | $1 \pm 0.0$ | $97 \pm 4.9$ | $100 \pm 0.0$ | $16 \pm 5.9$ |
| LSTM (forced) | Pair 1 | $100 \pm 0.0$ | $100 \pm 0.0$ | $25 \pm 6.1$ | $1 \pm 0.1$ | $76 \pm 10.0$ |
|  | Pair 2 | $100 \pm 0.1$ | $1 \pm 0.1$ | $94 \pm 4.2$ | $100 \pm 0.0$ | $42 \pm 14.7$ |

Table 3: Double-addition task: accuracy [%] of LSTMs and FNN on the two pairs. In case of LSTM (forced) only one input is presented at a time (to prevent interference). The header shows on which pair the mask was trained on. ¬ denotes an inverted mask for the hidden layers, while the regular mask (for the full task) is applied to the input and output layers. For further details please refer to Sec. C.5.1

of the *previous task* are initialized to 2 (corresponding to $P \approx 0.88$), the logits for newly initialized weights to either 0 ($P = 0.5$, Fig. 14a) or -1 ($P \approx 0.27$, Fig. 14b). Compared to Fig. 3, the sharing is significantly increased.

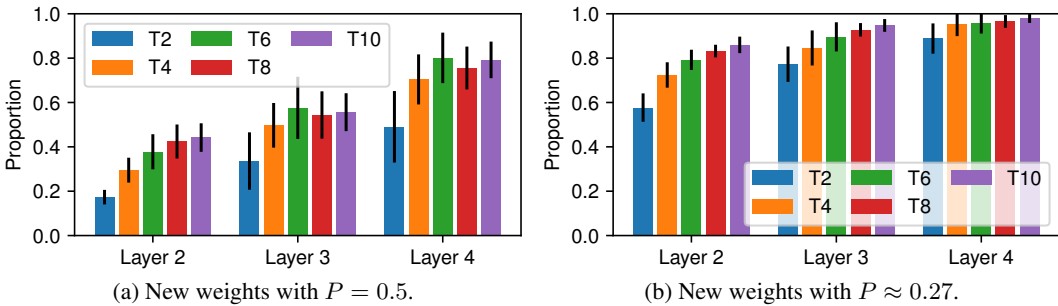

(a) New weights with $P = 0.5$.      (b) New weights with $P \approx 0.27$.

Figure 14: Proportion of weights shared per layer after every second task on permuted MNIST, for a network with masks initialized to prefer reusing the old weights. Old weights are sampled with $P \approx 0.88$ probability. Each task corresponds to a permutation. Decreasing the probability of new weights forces increased sharing.

## C.7   EXPERIMENTS ON ALGORITHMIC TASKS

### C.7.1   SCAN EXPERIMENTS

In preliminary experiments we observed that the full-size word embeddings used for the baseline in Lake & Baroni (2018) yielded many possible redundant input-to-hidden weight configurations that have a greatly reduced probability of being sampled. This caused the input-to-hidden layer to be removed by the thresholding procedure. Therefore, we appropriately reduced the size of word embeddings to 16 (note that SCAN has only 13 input and 6 output tokens and they are not shared). When using the reduced embedding we do not suffer from the aforementioned problems. Teacher forcing was used for each batch with $50\%$ probability.

The Transformer network is based on PyTorch's internal implementation, with modifications needed to apply multiple masks more effectively. We use $d_{model} = 100$, inner-layer dimensionality of $d_{ff} = 200$, $h = 4$ heads, and 3 layers both in the encoder and decoder. The network is always trained with teacher forcing. An end token is applied to the end of each input sequence, and decoding starts with a start token. The sinusoidal positional embedding (Vaswani et al., 2017) is applied to the inputs of the transformer in both encoder and decoder.

The training procedure uses a batch size of 256 and a gradient clipping of 5. The mask learning rate is $10^{-2}$ and we use $\beta = 3 * 10^{-5}$ for the LSTM experiments and $\beta = 10^{-3}$ for the Transformers.

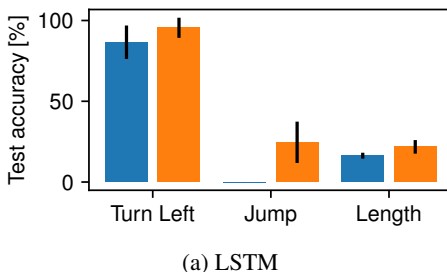
(a) LSTM

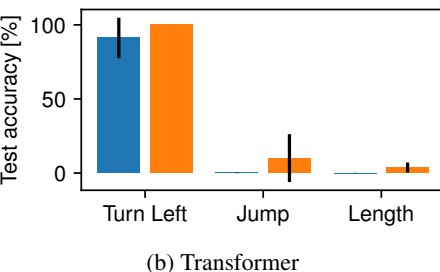
(b) Transformer

Figure 15: The networks' performance when it is directly trained and tested on the splits indicated on the $x$-axis, without masking. This is the standard setup from Lake & Baroni (2018). Performance when the network is trained on IID split, then masks are trained on train split indicated on the $x$-axis. It can be seen that although training on the IID set helps compared to the basic setup, the network still needs task-specific weights, which hurt performance when removed by the masks.

We train the networks for 25k steps without masks before freezing and for another 25k steps for each mask training phase.

We train the network weights on the IID dataset (the "simple" split), and only the masks on the rest of the data splits. Figure 15 shows that this process marginally improves the performance of all splits, compared to when the network is directly trained on the corresponding train split. However, the performance degrades significantly when removing weights that are unnecessary for the given training split. This allows us to conclude that the learned solution requires tasks-specific weights. Notice that the remaining weights still have significantly better performance than training the network solely on the training set of the given split without masking.

The word embeddings are excluded from the masking process and remain unmodified after initial training to keep the learned word representations unchanged.

### C.7.2 Experiments on the Mathematics Dataset

The Mathematics Dataset (Saxton et al., 2019) is a dataset intended to test the mathematical reasoning skills of NNs. It consists of 56 tasks from different areas of mathematics, on high school-level. Each task consists of a train set divided into 3 levels of difficulty (easy, medium, and hard) and an IID test set. All questions and answers are provided only in a human-readable text format (see examples in Fig. 16).

We train the network on individual tasks, in contrast to the method of Saxton et al. (2019), where all tasks are trained together. The main reason for this is to save computation and to prevent possible interference between the tasks. We chose five different tasks to analyze, based on their difficulty: the chosen tasks should have good performance without masking but should be nontrivial. Thus we choose "arithmetic: add_or_sub", "algebra: linear_1d", "calculus: differentiate", "comparison: sort" and "polynomials: collect". Note that the performance of our network might be less than reported in Saxton et al. (2019), since no transfer between tasks is possible, and we train for significantly fewer iterations because of limited computational resources.

We split the official easy, medium, and hard *train* sets to obtain new train and validation sets for each difficulty level. We randomly choose 10k samples for the new validation set; the rest is used as the new train set. We filter for repetitions, making sure that no sample appears twice. This way, we get a train and validation set for each difficulty level. We ignore the official test sets because of the missing distinction in difficulty. This treatment is needed because we want to be able to train the network on all difficulty levels but also the masks only on the easy split. Additionally, we want to evaluate its performance on the hard difficulty. In this way, we are able to determine whether specific weights are needed exclusively for performing the hard split. Note that the same rules govern the samples in all sets.

What is the difference between 1801791.2422 and −0.7?
1801791.9422

Solve −719∗o + 3179∗o + 135275 = −628∗o − 777∗o for o.
−35

What is the derivative of 30595∗j∗∗4 + 254∗j∗∗3 + 1559873 wrt j?
122380∗j∗∗3 + 762∗j∗∗2

Sort −3/5, −1355.6, 703, 2, −2/3 in ascending order.
−1355.6, −2/3, −3/5, 2, 703

Collect the terms in −26∗v − 67 + 29∗v + 12∗v − 3 − 155.
15∗v − 225

Figure 16: Examples from Mathematics Dataset. One sample for every task we use.

First, we train the network on all difficulty levels (easy, medium, and hard). Then we freeze its weights. Next, we train masks on the easy split and test on the hard split. If this results in a performance drop, then this indicates that the network requires a separate set of weights for different difficulty levels, which is undesirable. Nonetheless, we observe precisely this behavior (Fig. 5), which confirms once more that NNs tend to violate $P_{reuse}$.

Interpreting the size of the drop is nontrivial due to how the easy, medium, and hard splits differ. The more difficult splits may include some samples from the easier splits, but never the other way around. This means that the hard test set's performance will be nonzero even if none of the hard samples are solved correctly. This behavior is inherent to the original dataset and can not be changed without regenerating it.

We use the Transformer (Vaswani et al., 2017) model from Saxton et al. (2019). It has a $d_{model} = 256$, inner-layer dimensionality of $d_{ff} = 512$, $h = 4$ heads. Both the encoder and decoder have 3 layers. The word embeddings of the encoder and decoder are shared, and the output layer is tied to the word embedding. The network is always trained with teacher forcing. We use the Adam optimizer with a learning rate of $10^{-4}$, $\epsilon = 10^{-9}$, $\beta_1 = 0.9$, $\beta_2 = 0.995$ and gradient clipping of 1. We use 8 masks samples for each batch. We found that some tasks require a linear learning rate warmup for 5k iterations at the beginning of network training in order to converge. No warmup is used for training the masks. Individual tasks use different hyperparameters, listed in Table 4. Batch sizes are chosen so that the experiments fit on a single GPU with 16Gb of VRAM (2 GPUs for "Poly. collect").

| Hyperparameter | Add or sub | Linear 1D | Differentiate | Sort | Poly. Collect |
|---|---|---|---|---|---|
| Batch size (net) | 256 | 512 | 128 | 256 | 128 |
| Batch size (mask) | 256 | 400 | 128 | 256 | 256 |
| Mask regularizer ($\beta$) | $2 * 10^{-5}$ | $10^{-6}$ | $10^{-5}$ | $3 * 10^{-6}$ | $10^{-6}$ |
| Training iters (net) | 30k | 200k | 40k | 30k | 200k |
| Training iters (masks) | 30k | 50k | 40k | 30k | 50k |
| Learning rate (masks) | 0.03 | 0.02 | 0.03 | 0.03 | 0.02 |
| Warmup steps | - | 5k | - | - | 5k |

Table 4: Hyperparameters for different tasks on the Mathematics Dataset

## C.8 CNN Experiments on CIFAR10

### C.8.1 Simple CNN

We use a learning rate of $10^{-3}$ and $\beta = 10^{-4}$. We train the network for 20k steps before freezing its weights and then use an additional 20k steps for training each of the masks, including the reference mask. See Table 5 for details regarding the architecture.

| Index | Operation | Inputs | Outputs | Kernel | Padding | Activation | Dropout |
|---|---|---|---|---|---|---|---|
| 1 | Conv | 3 | 32 | 3x3 | 1 | ReLU | - |
| 2 | Max pooling | 32 | 32 | 2x2 | 0 | - | - |
| 3 | Conv | 32 | 64 | 3x3 | 1 | ReLU | - |
| 4 | Max pooling | 64 | 64 | 2x2 | 0 | - | - |
| 5 | Conv | 64 | 128 | 3x3 | 1 | ReLU | 0.25 |
| 6 | Max pooling | 128 | 128 | 2x2 | 0 | - | - |
| 7 | Conv | 128 | 256 | 3x3 | 1 | ReLU | 0.5 |
| 8 | Spatial average | 256 | 256 | - | - | - | - |
| 6 | Feedforward | 256 | 10 | - | - | Softmax | - |

Table 5: Architecture of the simple CNN used for CIFAR 10 experiments

Fig. 18 shows the confusion matrix difference for all classes of CIFAR 10. The most surprising observation is that the decrease in performance for each of the classes is substantial, ranging from 40 to 60%. This shows the heavy reliance on class-exclusive features.

Analyzing confusion matrix differences yields interesting insights. "Airplane" is confused with"bird" and "ship", which is likely due to having a similar blue background. Classes "cat" and "dog" tend to be confused with each other—removing exclusive feature detectors for one improves the performance of the other. "Truck" and "car" are highly related, likely due to the similarities in terms of shape, such as having tires, and similar backgrounds, such as the road.

### C.8.2 Simple CNN Without Dropout

The CNN architecture used for experiments in Section 5 uses dropout, as shown in Table 5. A natural question to ask is how this affects the modularity of the resulting network. Figure 17 indicates that, as expected, removing dropout results in a few percent of performance loss. When comparing Figures 18 and 19 it can be seen that adding dropout causes a higher degradation in the class performance when the class-exclusive feature detectors are removed (roughly 30%-40% higher drop per class). This indicates that network with dropout depends more on class-specific modules, which is in line with findings presented in Filan et al. (2020).

### C.8.3 ResNet-110

To demonstrate that these behaviors apply to more complex models, we train a ResNet-110 (He et al., 2016) model which achieves competitive 93% validation accuracy following `https://github.com/bearpaw/pytorch-classification`. The network is built from non-bottleneck blocks ("BasicBlocks", Fig. 5, left in Vaswani et al. (2017)). It is trained with SGD using a weight decay of $10^{-4}$, batch size of 128 and a starting learning rate of $0.1$. The learning rate is divided by 10 at iterations $32\,000$ and $48\,000$ (corresponding roughly to epoch $81$ and $122$). The network is trained for $64\,000$ iterations (164 epochs). Data augmentation of random horizontal flipping and random crop (with padding $4$ and output size of 32x32) is used. Masks are trained with Adam, batch size of 256, learning rate of $0.03$, $\beta = 2 * 10^{-5}$, for $30\,000$ iterations each. Gradient clipping is not applied during the initial stage of training the weights, but the usual clipping to norm of $1.0$ is applied when training the masks.

As Figures 6 and 20 show, the performance drop per class is even more dramatic than in the simple CNN case, reaching almost 100%.

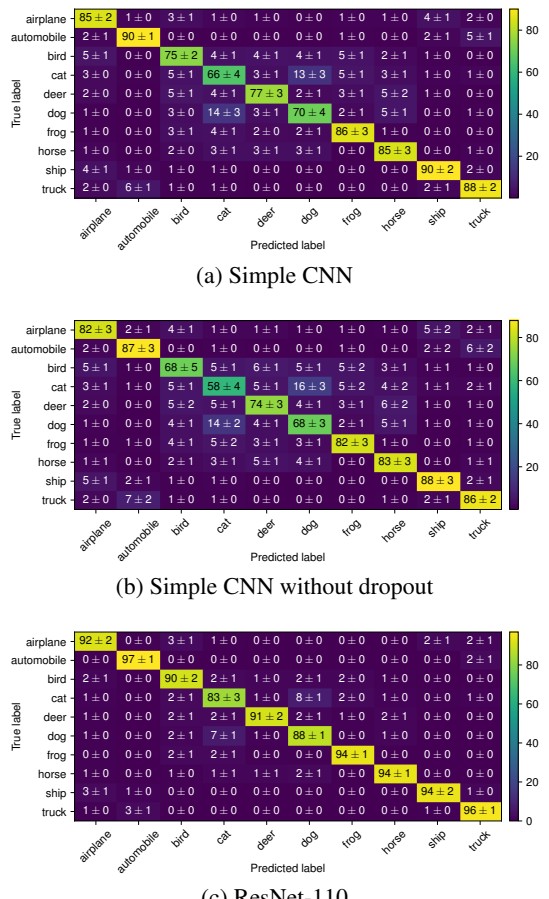

(a) Simple CNN

(b) Simple CNN without dropout

(c) ResNet-110

Figure 17: Confusion matrix on CIFAR10 with masks trained on all classes. It can be seen that performance without dropout is a few percent lower, as expected. ResNet-110 has a significantly better performance in all classes.

Inspecting the confusion matrix differences of different architectures as seen in Figures 18, 19 and 20 highlight their similarity. This suggests that the interdependence between classes previously observed is mostly data driven an independent of the actual network architecture.

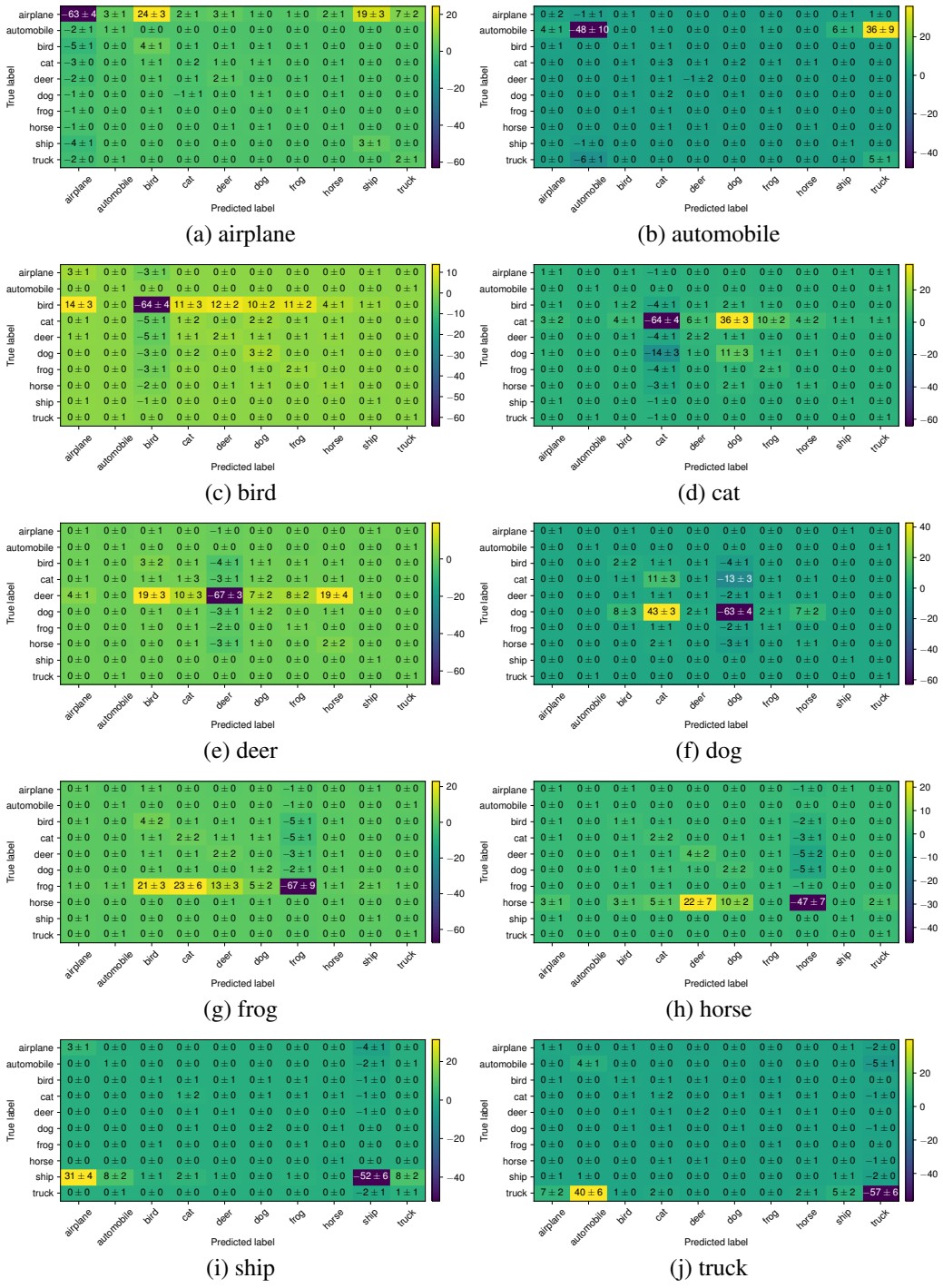

Figure 18: Simple CNN: The change in confusion matrix for all CIFAR10 classes, when class indicated by the caption, is removed.

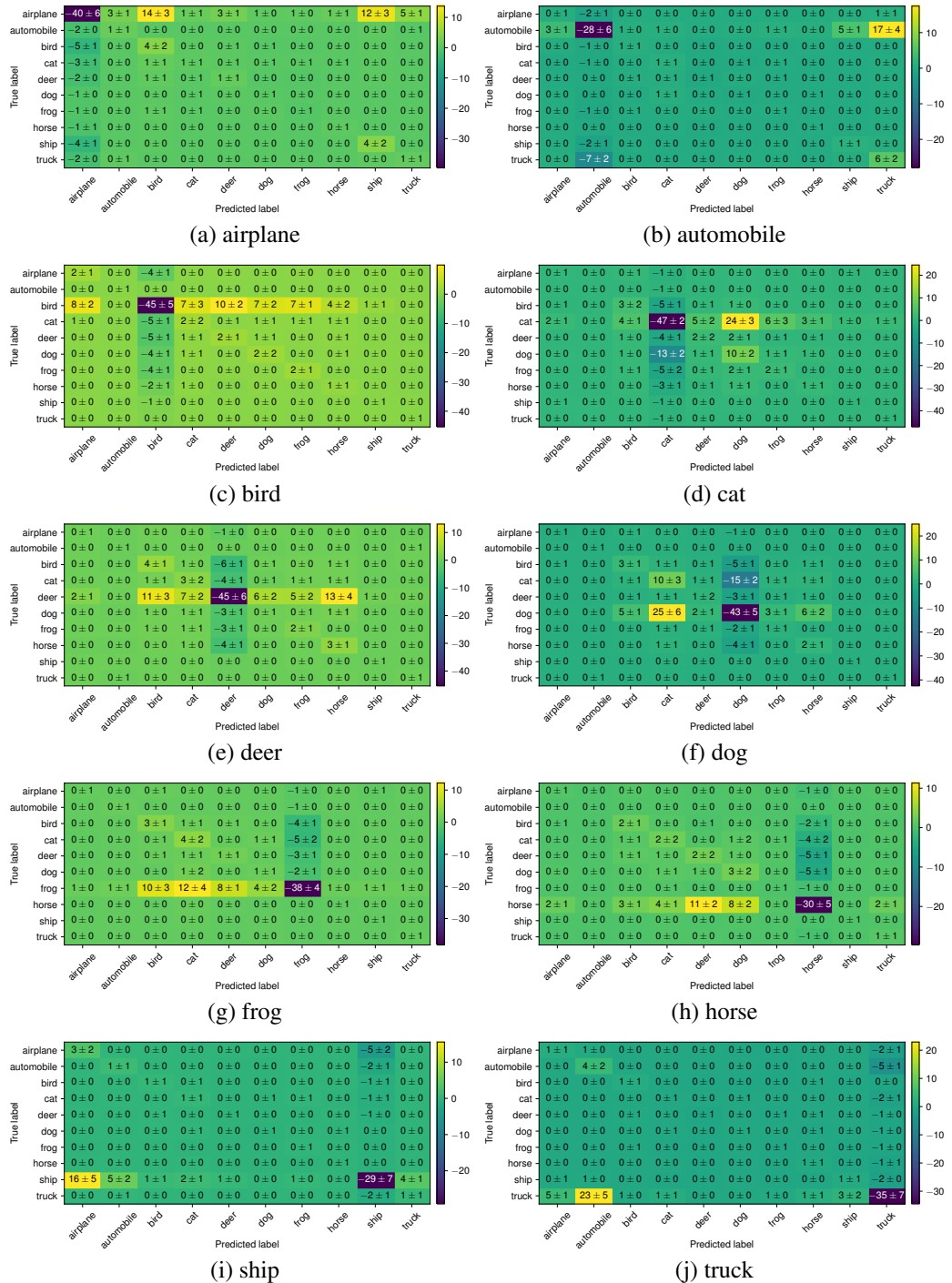

Figure 19: Simple CNN without dropout: The change in confusion matrix for all CIFAR10 classes, when class indicated by the caption, is removed. The network has the same architecture as Table 5, but without the dropout layers. The performance drop is reduced by roughly 30%-40% compared to the same architecture with dropout (Fig. 18).

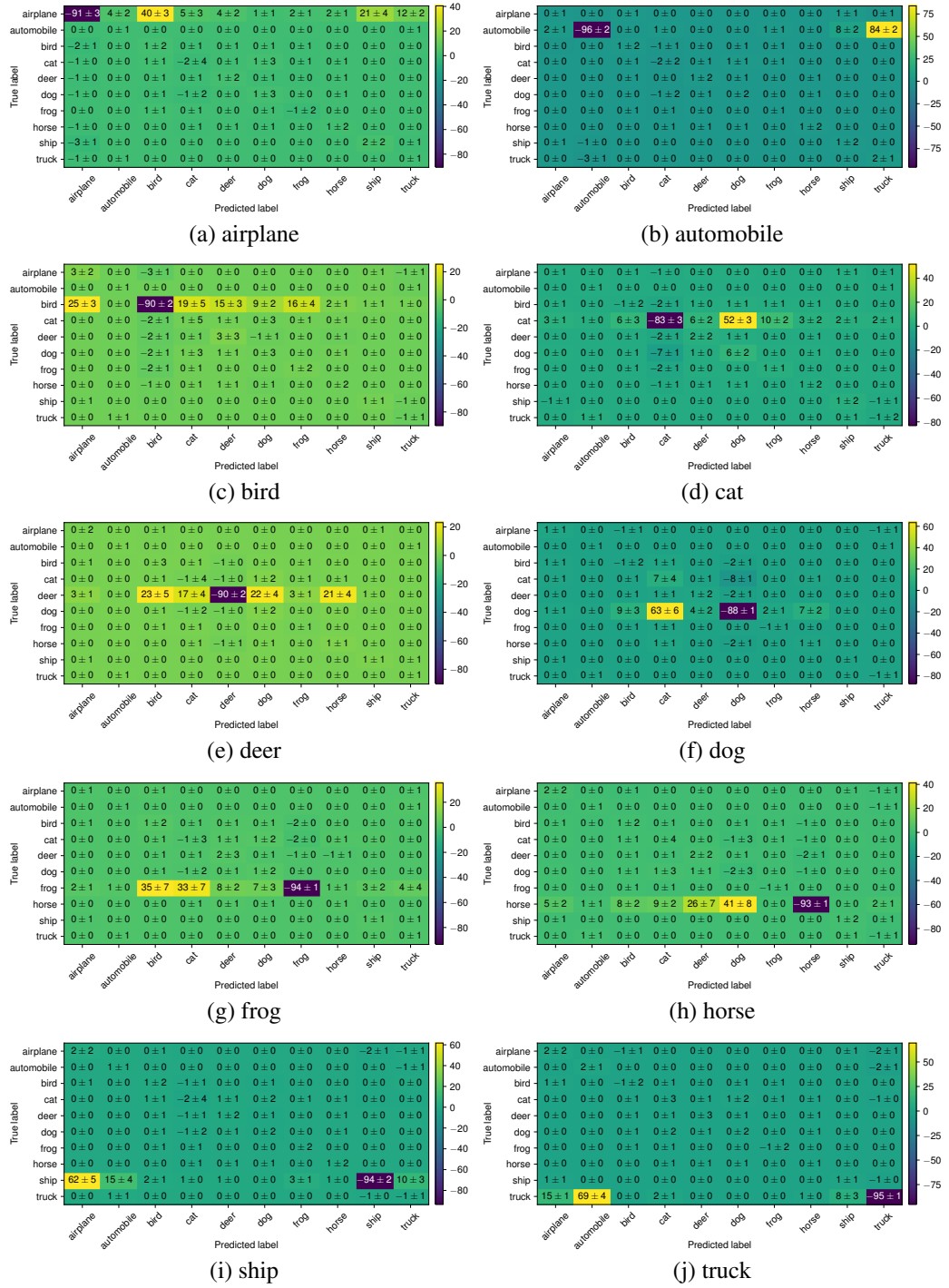

Figure 20: ResNet-110: The change in confusion matrix for all CIFAR10 classes, when class indicated by the caption, is removed.

