# OpenReview forum: "Are Neural Nets Modular? Inspecting Functional Modularity Through Differentiable Weight Masks"
_ICLR.cc/2021/Conference — ICLR 2021 Poster_

### Official Review · AnonReviewer4 · 2020-10-25
**Interesting exploration of modularity in Neural Networks, although some conclusions not fully justified**

**Rating:** 8
**Confidence:** 3

**Review:**

This paper studies weight modularity in neural networks (NNs). In particular, given a NN trained to perform a task, a subset of weights are identified which in isolation perform well on a subtask of the original task. Such subsets are inspected to understand the extent to which they are specialized or reused across different subtasks of the original task. To identify subtask specific weights, a mask is learned that minimizes loss over a subtask when applied to the original NN's frozen weights. This process is carried out using gradient based optimization techniques (Adam). Extensive experiments are performed across various datasets and architectures. The paper concludes that while NNs seem to exhibit module specialization, they fail to exhibit reuse.

### Strengths

1. Understanding modularity within NNs and its relation to failures of systematic generalization is an important research direction.
2. The precise masking method proposed here is novel to the best of my knowledge.
3. The functional view of modularity has clear advantages over clustering based approaches.
4. I believe this work could be extended in interesting ways. For example, it seems like the methods presented here could naturally be extended to encourage modularity, rather than measure it.

### Weaknesses

1. The paper addresses a well known issue, that NNs often fail to generalize systematically. Why this occurs and potential solutions are left unaddressed. While the paper does put forward a hypothesis regarding why, that the difficulty of learning routing operation in NNs may lead current approaches to fail to discover solutions which generalize systematically, little evidence is provided to support this conclusion. As such, I believe claims such as, "our approach can also be applied in these settings to provide additional insight regarding the underlying reason for the observed failure at systematic generalization" should be relaxed, or clarified if there is evidence for this hypothesis presented that I am failing to connect.

2. How do we know the problem is with the full model, rather than the mask? Recent literature (cited in the paper) demonstrates that performant models can be obtained even when severe weight constraints are imposed [1, 2]. Indeed, [2] demonstrates masking alone is sufficient to adapt a NN to a completely new task. Given these observations, I'm concerned about the validity of assuming the mask optimization procedure identifies an independent functional module in the context of the computation performed by the full model. It seems plausible that the mask performs well on the subtask given the fixed model weights, but is unrelated to the function of the full model. If this was the case, I believe a number of the conclusions in the paper would not be justified.

  The paper does provide some evidence this may not be occurring. For example, the double-addition experiment (Figure 2) shows that inverted masks perform well on the other pair. This suggests the mask and inverse mask may correspond to distinct functionality in the full model. However, Figure 10 tells a different story, where inverse masks do not perform well on the other task. Is this due to the shared IO in the addition/multiplication task? Is there any relation between the addition mask with shared weights removed and the inverted multiplication mask with shared weights removed?

   Ideally, the paper would provide more evidence to support this assumption. Are units correlated in the masked and unmasked model? Could the analysis here be supported by explicitly constructing a model that exhibits reuse, and verifying it can be identified?

3. The paper argues in several places that a weight level analysis is necessary for reasoning about functional modularity, e.g., "without considering the contribution of individual weights it is not possible to reason about *functional* modularity." I believe this statement conflates two separate issues; (1) if modules should be defined in terms of units or weights, and (2) whether modules should be defined by functionality (as done in this work) or using techniques such as clustering. On the latter point, I agree with the statements in the paper and believe the functional approach is more closely aligned with intuitive notions of what is meant by modularity. However, I don't believe a weight vs unit level perspective is necessarily mutually exclusive. For example, consider a simple affine layer $h = f(Wx)$. Defining a module as a subset of elements of $h$ would be equivalent to defining a module as a subset of rows in $W$.

4. Figures 2, 10, 14 would be much clearer if presented in a tabular format. I had to write out the results this way myself to aid understanding.

5. I found the use of a non-symmetric sharing metric extremely confusing. I spent a while pondering over Figure 1 trying to figure out what the weights were shared with since both tasks are represented in the figure. I believe this issue could be easily remedied, for example, by using something like IoU (Jaccard index). The paper uses this metric in Appendix B.1 to determine the amount of sharing between different masks trained on the same network and task.

6. Permuted MNIST Experiment

   1. The paper states "it suffices to re-train a new first layer to undo the permutation so that later layers can be reused." This makes sense. However, given that the procedure is "freeze the occupied weights," lower level weights cannot perform this simple permutation by construction since they are frozen (I've assumed that occupied means a non-zero mask value).
   2. Why was the choice made to "train masks and weights simultaneously" in this setting? This introduces an additional source of variation from previous experiments, and I'm unsure what the benefit is.

7. SCAN experiments

   1. The paper states "if the masking process removes any important weights, then the solution is pattern-recognition like instead of being based on reusable rules, confirming explanation." This is related to point (2) above, and I'm not convinced this is the only possible conclusion. While the mask has identified a subnetwork whose solution is pattern-recognition like, I'm not sure that the logical jump to conclude the model as a whole performs identically is sound.
   2. Regarding the output weight analysis, I'm also unsure why it should be the case that the final layer is sufficiently powerful for unbinding bound variables. Isn't it also possible that the mask learns to ignore those weights because it is not important for the task it is trained on?

### Recommendation

I recommend acceptance. Although I have several concerns with the paper, I found the ideas and analysis presented very interesting. I believe the community would benefit from further discussion, scrutiny, and exploration of the ideas presented.

### Post Author Response Period Update

Most of my concerns have been addressed by the additional experiments and updated language in the latest revision. I believe the techniques and analysis presented here for assessing reuse could be an important step between observations and explanations for the failure of NNs to generalize systematically. I have raised my score accordingly.

### Minor Issues

1. Appendix C.2: "We always check the **choosen** hyperparameters" => "We always check the **chosen** hyperparameters"

### References

1. Arun Mallya, Dillon Davis, and Svetlana Lazebnik. Piggyback: Adapting a single network to multiple tasks by learning to mask weights. In Proceedings of the European Conference on Computer Vision (ECCV), pp. 67–82, 2018.
2. Adam Gaier and David Ha. Weight agnostic neural networks. In Advances in Neural Information Processing Systems, pp. 5364–5378, 2019.

---

> ### Author Response · Authors · 2020-11-17
> **Response to reviewer 4 with comments on improvements**
>
> Thank you for taking the time to review our work in detail and provide valuable feedback. We were very impressed with the thoroughness of your review and really appreciate the effort that you put into reviewing our work, which allows us to make several improvements. Below are the individual responses to the many specific queries and comments:
>
> 1.  Thank you for pointing this out, this is mostly an oversight on our part. We will relax the claims about the insights to better reflect the fact that we cannot provide definitive evidence to support our hypothesis in some cases.
>
> 2.  > “How do we know the problem is with the full model, rather than the mask?.. ”
>
>   This is a good point and although we strongly believe that the problems are on the model side (as is also evident from the inverse masking results as you point out), we have devised an additional experiment to demonstrate that the masking process does not change the original function performed by the network. In particular, we will attempt to evaluate a network where only the first half of the network is masked, while the second is not. If the network works like this, it should mean that the activations produced by the first, unmasked, half are compatible with the second, masked half. Alternatively, we could calculate additional statistics of the activations, or plot the activations of the same sample with and without masking, to provide further evidence to this point.
>
>   > “However, Figure 10 tells a different story, where inverse masks do not perform well on the other task. Is this due to the shared IO in the addition/multiplication task?”
>
>   Figure 10 tells a different story for multiple reasons. First, the IO is shared as you also noted. Second, the separation of the hidden layers is not perfect. We believe that the sharing of weights is explained more by sharing the module’s inputs/outputs than by the performed function in this case. Note, however that Figure 10 is mostly there for symmetry with the double-addition experiments, which provides additional insight.
>
>   > ”Is there any relation between the addition mask with shared weights removed and the inverted multiplication mask with shared weights removed?”
>
>   The addition mask with the shared weights removed is a subset of the inverted multiplication mask with the shared weights removed. Both of them would perform poorly because the shared weights are important for the task.
>
>   An additional experiment that we could perform is to modify the inverse-mask experiment on the addition/multiplication task such that the inverted mask will be unioned together with the shared weights. This would just remove the weights exclusive for one operation. The performance on that operation should decrease, but the inverse mask should perform well on the complementary task. Personally, we don’t expect that this will provide much additional insight, but if you think that this is valuable then we are happy to run this. Please let us know in this case.
>
> 3. The paper argues for both (1) and (2), but we are also specifically advocating for (1) when mentioning weight-level analysis. We are not arguing that unit-wise analysis necessarily gives a wrong result, but rather that it is a special case of the weight-level analysis that is conducted here, and in practice probably not enough to draw meaningful conclusions. The provided example with the affine layer can demonstrate this easily: imagine x has 4k units, while h has k. X can be partitioned to 4 equal parts, a,b,c,d. The task is to perform either h = a  |  b or h = c & d, depending on which inputs are present. This can be done with a single affine layer. Now if we conduct a unit-wise analysis on h, all units will equally take part in both | and & operations. In contrast, a single row of W can be divided into 2 parts, where the first half performs the | and the second the &. This is why we are arguing explicitly for (2) in this case.
>
> 4. Thank you for the feedback. We are currently looking to improve this visualization and will let you know once the paper is updated with this. We will also add tabular results to the appendix.
>
> 5. Using IoU is a good idea for these figures, thank you for the suggestion. We will change the corresponding figures in the main paper accordingly.

---

> > ### Author Response · Authors · 2020-11-17
> > **Response to reviewer 4 with comments on improvements, part 2**
> >
> > 6. Permuted MNIST Experiment
> >    1. That is a great point, which we had not considered and is an oversight on our part. However, we don’t believe that the results for these experiments are invalidated, since the first layer is extremely sparse (meaning that the classification is done only based on few pixels). Hence, currently such conflicts are highly unlikely as most of the weights are freely available. Nonetheless, to avoid even the slightest chance of such conflict, we will re-run these experiments with the first layer fully reinitialized, without freezing any weights. We will notify you once we have updated the results in the paper.
> >    2. For the variants where we bias the network towards reusing the old weights artificially, the mask probabilities are used to inject the preference of weights in the network. If the network is trained in 2 stages, the masks would not have any effect on the weights in this case.
> >
> > 7. SCAN experiments
> >    1. Performing the half-masking experiments proposed in 2. should make this point much stronger, we believe.
> >    2. In case variable binding is performed, unbinding should be the last operation, which should map the result to the output token. While we can not know whether the last layer is sufficient for unbinding, it should definitely be part of it (or else it is redundant) and not perform further operations. Hence, it should not have an excessive number of weights that are dependent on the specific compositions of the elementary operations the input was built from, but should mostly depend on what the output is, which is not the case.

---

> > > ### Comment · AnonReviewer4 · 2020-11-20
> > > **Most concerns resolved**
> > >
> > > *Note: I have added an additional comment to Reviewer 1's thread discussing the validity of some conclusions drawn in the paper, I address other points from the author's response below*
> > >
> > > > In particular, we will attempt to evaluate a network where only the first half of the network is masked, while the second is not
> > >
> > > To clarify, this would only be done after the mask is trained, correct? In general, I agree with the author's that this would provide further evidence to support the interpretation of the masks presented here. Would it make sense to evaluate the reverse as well (first half unmasked, second half masked)?
> > >
> > > I would still like to see a sanity check where a model featuring reuse can be correctly identified, even if the model has to be explicitly constructed. However, I understand this may be difficult in the time remaining.
> > >
> > > > We are not arguing that unit-wise analysis necessarily gives a wrong result, but rather that it is a special case of the weight-level analysis that is conducted here
> > >
> > > I agree with this sentiment. This could be made more clear in the paper (or I missed it).
> > >
> > > > For the variants where we bias the network towards reusing the old weights artificially, the mask probabilities are used to inject the preference of weights in the network. If the network is trained in 2 stages, the masks would not have any effect on the weights in this case.
> > >
> > > I may have been unclear in my initial comment. In the other experiments, the setup is always learn on all task, then learn a mask on a subtask. The reason for the deviation in the MNIST experiment appears to be to see if you can encourage weight sharing. Is this correct? If so, this should probably be stated earlier in Section 3.3. Furthermore, I'd expect incorporating an explicit term in the loss to encourage mask sharing to be more effective than initialization alone. Did the authors consider this?
> > >
> > > > While we can not know whether the last layer is sufficient for unbinding, it should definitely be part of it (or else it is redundant) and not perform further operations.
> > >
> > > I'm not convinced by this. For example, it seems equally likely that the last layer is responsible for mapping an already unbound internal representation to the correct output.

---

> > > > ### Author Response · Authors · 2020-11-23
> > > > **Response to reviewer 4**
> > > >
> > > > > “To clarify, this would only be done after the mask is trained, correct? In general, I agree with the author's that this would provide further evidence to support the interpretation of the masks presented here. Would it make sense to evaluate the reverse as well (first half unmasked, second half masked)?”
> > > >
> > > > Yes, correct, it is done after all the training is done (both weights and masks). According to your suggestion, we also did the reverse, and we report both in Appendix B.2.
> > > >
> > > > > “I would still like to see a sanity check where a model featuring reuse can be correctly identified, even if the model has to be explicitly constructed. However, I understand this may be difficult in the time remaining.”
> > > >
> > > > We ran additional experiments on the double-addition task, where after the training phase we copy the weights in the input/output layers from one pair to the other (Appendix C.1). This makes the two pairs indistinguishable from the perspective of the hidden layers. Our method discovers almost perfect sharing in this case, confirming that it is capable of discovering sharing.
> > > >
> > > > > “I agree with this sentiment. This could be made more clear in the paper (or I missed it).”
> > > >
> > > > We believe that we have clarified this in the updated paper.
> > > >
> > > > > “The reason for the deviation in the MNIST experiment appears to be to see if you can encourage weight sharing. Is this correct? If so, this should probably be stated earlier in Section 3.3.”
> > > >
> > > > Yes that is correct. We believe that we have clarified this in the updated paper.
> > > >
> > > > > “Furthermore, I'd expect incorporating an explicit term in the loss to encourage mask sharing to be more effective than initialization alone. Did the authors consider this?”
> > > >
> > > > Following your earlier suggestions, we changed the transfer learning experiments such that the first layer is never frozen and its weights are reset for each permutation. This ensures that the first layer is never a bottleneck, and the permutation can always be undone. This did not change our main conclusion (no change is noticeable), but it changes the results where we force sharing. With these modifications, we can force the network to share by purely the mask initialization, so no such regularization is necessary any longer.
> > > >
> > > > > “I'm not convinced by this. For example, it seems equally likely that the last layer is responsible for mapping an already unbound internal representation to the correct output.”
> > > >
> > > > Yes, we completely agree with that. But if it is mapping the unbound internal representation to the output, then it should not have combination-specific weights, we argue. Rather, its weights should just depend on the output token in this case. In contrast, we observe weights in the last layer, which are combination-specific.

---

### Official Review · AnonReviewer1 · 2020-10-27
**In its current version disqualified for formal reasons; Lacks discussion of several areas of relevant literature; Main result cannot convince**

**Rating:** 6
**Confidence:** 5

**Review:**


In its current form, I feel that the paper should be disqualified because it contains some results essential to its claims in the appendix (referenced in the second paragraph, page 5). However, this can be easily addressed - thus my full review below:

Summary of the paper:
The paper aims to analyze if and how neural networks learn modular representations. Modularity under the paper's definition means that the network learns representations that (1) specialize (using different modules for different functions) and that (2) compose in a re-usable fashion - i.e., that functions are used in diverse tasks.
To do so, the authors train different probabistic masks (using a Gumbel-Sigmoid) on the weights of a neural network over a series of different tasks. They incentivize these masks to be sparse by regularizing the number of "active" elements in a mask.
They then compare masks learned for different tasks (including simple arithmetic tasks, and permuted MNIST), and then compare the usage of the parameter masks over different tasks. They find that NNs learn to specialize only, without reuse.
The paper continues by postulating that the reason is either that the network learned "bad" representations, or that the network did not learn the correct composition. They argue that results on the SCAN dataset show that the representations are of a sufficient quality, concluding that the network did not learn the correct composition.

Commentary:
The question the paper tries to answer is very relevant, on two levels. The first is that we do not understand how an NN learns sufficiently well. The second is that compositional modularity is a highly desireable property, and it is important to know if neural network exhibit it.

Strengths:
- The paper does some interesting analysis
- In general, the paper is well-written, and easily understood.

Unfortunately, the paper suffers from major drawbacks:
- (this is a minor point I'm putting here to facilitate my discussion below) The paper is not positioning itself correctly in the literature, thereby using confusing terminology
While modularity is not a main focus in neural network research, there exists some meaningful research that has established some terminology. In essence, the paper talks about compositional modularity (and combinatorial generalization), but does not use this terminology. This makes the paper a little difficult to follow, if one is familiar with this literature. Within this terminology, specialize would be called "modularize" and Preuse would be called "compose", a terminology I will use in the following.
- The conclusions are not convincing
The core argument in the paper is that neural networks fail to modularize because they either learn insufficient representations, or because they fail to learn to compose (to learn the "algorithm" required to utilize the modules correctly). Because the network learns re-usable representations, they modularize, and must thus fail to compose.
While this is probably true, I cannot help to feel a little underwhelmed. It is well known that neural networks are overparameterized (see the "lottery ticket" literature). For this analysis, this means that it appears to be easier for the model to re-learn using the available capacity than to re-use the existing modules.
This is not particularly surpsising either, because this is, at its core, overfitting: the model does not generalize (which, in effect, is a "softer" way to re-use), but instead learns something akint to a separate function for different inputs.
What the paper does not investigate is why a neural network not re-uses capabilities, even those should be a good fit for a problem. This would be a really interesting analysis, one which I would very strongly argue for acceptance in any venue.
- The novelty of some parts is overstated
This is particularly true for using binary masks for multi-task like learning. See "Bengio, E., Bacon, P. L., Pineau, J., & Precup, D. (2015). Conditional computation in neural networks for faster models. arXiv preprint arXiv:1511.06297."
Additionally, the paper does not sufficiently relate their insights to the (cited) work around inducing modularity and compositionaly in networks, some of which does already come to similar results.

---

> ### Author Response · Authors · 2020-11-17
> **Response to reviewer 1 with comments on improvements**
>
> Thank you for taking the time to review our work in detail and provide valuable feedback. Although we were initially glad to see that you found the presented analysis interesting, we were disappointed at receiving a final rating of a 3.
>
> Ignoring your comments about results in the appendix (which, as you also state, are easily addressed) and terminology (which you state is minor) for now, it appears that there are two reasons that led you to this score:
>
> 1. You argue that “The conclusions are not convincing”, since “it is well known at neural networks are overparameterized”, and that therefore the presented results are “not surprising”. You suggest that we investigate an alternative research question: “why a neural network not re-uses capabilities, even those should be a good fit for a problem”.
>
> 2. You argue that the “novelty of some parts is overstated”, which you believe to be “particularly true for the binary masks for multi-task learning”. Additionally, you feel that “the paper does not sufficiently relate their insights to the (cited) work around inducing modularity and compositionality in networks, some of which does already come to similar results.”
>
> Regarding (1), we would like to emphasize the distinction between results not being “convincing” and them being “expected”. Although you state that our analysis is not “convincing”, you yourself point out how our findings are “probably true” and do not offer any concrete remarks or evidence that suggest that the presented analysis is not thorough or even invalid. As to whether the results can be “expected”, clearly this is debatable. When we started this work a year ago it was far from obvious that these results would follow from the “lottery ticket hypothesis” and arguably it still isn’t without first considering the evidence presented in this work. Indeed, to the best of our knowledge there has been no prior analysis that studies such effects in neural networks, which we argue makes this work a valid contribution, irrespective of whether, in hindsight, this may have been expected.
>
> In judging the significance of our results we therefore ask that you also consider the comments from the other three reviewers, none of which suggested that these results are obvious or logically follow, which we argue indicates that our results are significant. Further, we argue that it is important to consider the understanding of the broader community (as opposed to expert reviewers) regarding these issues in neural networks when judging the significance of our findings. It is our hope that you will reconsider.
>
> Regarding (2), it was never our intention to claim that multi-task learning with binary masks is a novel approach that we propose. The sentence “Typical approaches revolve around freezing used weights when a new task is added (Fernando et al., 2017; Mallya & Lazebnik, 2018; Golkar et al., 2019).” was meant to clarify this. However, we now realize that it may not be obvious to the reader that this implies that these methods use a form of masking. This is an oversight on our part and we thank the reviewer for pointing this out. The next revision will clarify this. We also thank the reviewer for pointing out Bengio et al., (2015), which we were unaware of and which we will include in the next revision as related work. More generally, regarding novelty for the multi-task setting, we note that our analysis of the behavior of the multi-task learner is the novel part, which is what this experiment contributes.
>
> Regarding the second part of (2), we believe that we have sufficiently compared our work to the many of the related works on modularity. To the best of our knowledge these works each focus on different aspects of this problem and do not not analyze modules based on their functionality, which is a defining characteristic of our work. If you disagree, could you please elaborate further on related work that we are not comparing to and that provides similar results?
>
> We will now provide individual responses to the remaining minor comments:
>
> > “In its current form, I feel that the paper should be disqualified because it contains some results essential to its claims in the appendix (referenced in the second paragraph, page 5)”
>
> The main results and conclusions from the second paragraph, page 5 are shown in Figure 2. Figures 13 and 14 are only referenced to support our claims. We agree that it would be better if we can present it in the main paper, however considering the page limit, this was challenging. We believe that with the extra page, we can do a better job at this.

---

> > ### Author Response · Authors · 2020-11-17
> > **Response to reviewer 1 with comments on improvements, part 2**
> >
> >
> > > “The paper is not positioning itself correctly in the literature, thereby using confusing terminology While modularity is not a main focus in neural network research, there exists some meaningful research that has established some terminology. In essence, the paper talks about compositional modularity (and combinatorial generalization), but does not use this terminology.”
> >
> > We believe that we are aware of a good part of the compositional generalization and modularity literature, but not one of the “compositional modularity” literature. We would appreciate it if you could point to some specific literature that references the terminology that you wish us to adopt.
> >
> > > “reuse would be called "compose"”
> >
> > Based on our current understanding (and lacking a reference to the terminology that you are referring to), we would argue that the meaning of “reuse” is very different from the meaning of “compose”. We understand “compose” as putting together the final solution by combining some elementary parts. This, in general, requires reusing but is also possible without reusing, just not very useful. On the other hand, we understand “reusing” as using the same module for multiple tasks. For example, there is nothing to compose in the double-addition experiments: they consist only of elementary operations. However, the adder can be reused.
> >
> > > “specialize would be called "modularize"”
> >
> > Similarly, we would here argue that their meanings do not overlap completely. To our understanding modularize implies that modules are formed in arbitrary ways, while specialize implies the modules are formed to respect some functional specialization. Based on your comments we will clarify “reuse” and “specialize” in the updated draft of the paper, although we would appreciate some clarification regarding the established terminology that you are referring to.
> >
> > > “What the paper does not investigate is why a neural network not re-uses capabilities, even those should be a good fit for a problem.”
> >
> > Indeed, this is a very different research direction compared to our current focus. Although we agree that it would be interesting to study this, it is unclear how this can be done beyond demonstrating better results on tasks that require such an approach (as is the focus of several other works). An advantage of our current research focus and methodology is that we are able to provide additional understanding beyond reporting improvement in accuracy as a result of applying masks in different ways at test-time.

---

> > > ### Comment · AnonReviewer1 · 2020-11-18
> > > **Some concerns addressed; however, major concerns remain**
> > >
> > > Thank you for your thorough response to my concerns.
> > >
> > > First, let me point out that Openreview allows to update revisions after the first review phase, and several other papers I am reviewing have done so in the last couple of weeks. The authors could have used this to address my editorial concern on the reference to the appendix.
> > >
> > > I am inclined to agree with the reviewers that my expectations to the novelty of the approach were maybe a little too high, and that their analysis has merit. I will increase my score based on this.
> > >
> > > However, I am still not convinced by the main argument in the paper - that "neural networks fail to modularize because they either learn insufficient representations, or because they fail to learn to compose". I gave this some more thought over the last couple of weeks, and I think that I can give a better description of my concern than the one I raised before (I apologize for bringing this argument up only now):
> > >
> > > The authors formulate their argument as "has to be A or B, it's not A, so it's B", with A being insufficient representations, and B being a failure to learn the algorithm to compose. The main flaw in using this argument in this context is that it only works if A and B are independent.
> > > I would go as far as to argue the opposite: representations and algorithm are inseparable when it comes to what an NN learns. In fact, it learns the exact combination of representation and algorithm that allows it to fit the train set, making me wonder if a general purpose NN can even learn such a decomposition if it is not architecturally motivated to do so. (One example that comes to mind is the face/noface picture classifier that just learns to pick up the background bokeh because face pictures have higher bokeh: the representation obviously contains something impacted by bokeh, and the rule can pick up on it)
> > > That also means that the failure has to do with representations, and with rules, as the authors state. However, the "either or" argument fails. This also allows me to re-phrase my previous statements about overparameterization and overfitting: the network overfits both the representations and the rules (but possibly only to each other - meaning that the rules cannot comsume arbitrary representations, and the representations require specific rules to learn). This means that for an overparameterized network, it is easier to re-learn for a new problem, because the existing rules/representations do not make sense in isolation.
> > >
> > > In combination, I therefore stand by my earlier assessment of rejection, because I still find the main argument of the paper not convincing. However, I truly look forward to seeing an updated version of this paper that tries to analyze why this happens a little more.

---

> > > > ### Author Response · Authors · 2020-11-19
> > > > **Response to reviewer 1**
> > > >
> > > > Thank you for engaging with us during the rebuttal period. Below is our response to your comments.
> > > >
> > > > > “First, let me point out that Openreview allows to update revisions after the first review phase, and several other papers I am reviewing have done so in the last couple of weeks. The authors could have used this to address my editorial concern on the reference to the appendix.”
> > > >
> > > > It is our understanding that the rebuttal period has only been under way for about 8-9 days or so (depending on your timezone). So far we have used this time to write detailed replies to each of the reviewers as well begin running new experiments. In this way, reviewers can already respond should they require further clarification, or disagree, etc.
> > > >
> > > > We will use much of the remaining time to draft a new version of the paper that incorporates all feedback, including yours. We are currently aiming for 1-2 larger updates to burden the reviewers as little as possible.
> > > >
> > > > > “However, I am still not convinced by the main argument in the paper - that "neural networks fail to modularize because they either learn insufficient representations, or because they fail to learn to compose"”
> > > >
> > > > This is not the main argument of the paper. The main argument is that neural networks resist weight sharing to implement shared functionality, and we present ample evidence that _directly_ supports this hypothesis (as opposed to the "has to be A or B, it's not A, so it's B" that you suggest).
> > > >
> > > > We believe our experiments on the SCAN and Math dataset may have led you to believe an alternative argument, where we use a somewhat similar argument, but we think you misunderstood our reasoning.
> > > >
> > > > We don’t claim that representations, as such, can be learned independently of the modular structure and neither investigate this. Rather, these experiments are aimed at answering a non-trivial question regarding the observed behavior (i.e. failure to generalize systematically) in prior work that analyzes systematic generalization, eg. SCAN [Lake & Baroni, 2018]. In those set-ups, the train set is systematically different from the test set. Sometimes this difference is extreme: for example in the “Add jump” split, the input “jump” is presented in complete isolation without combining them with other tokens. Therefore, it remains unclear why the network performs poorly.
> > > >
> > > > One hypothesis is that this is purely because the train set is too different from the test set and it is insufficient to learn to represent the isolated token (“jump”)  in a way similar to the well performing ones (“walk”, “run”, etc). In this case the network essentially fails to “make an analogy” between the good and bad performing tokens. For example in SCAN, it could fail to represent “jump” in a way compatible with “walk”, “run”, etc, even though each of these can be composed with other tokens in exactly the same way. This is what we call the problem of not learning a good representation.
> > > >
> > > > An alternative hypothesis is that there is a deeper issue beyond incorrectly mapping inputs that should be handled similarly, which we investigate. By training the network weights on IID data, i.e. the network is provided an exhaustive number of combinations, we allow it to learn to perform well on all splits. This rules out the potential problem of not having enough data to realize that similar tokens should be handled similarly (as per the previous hypothesis), and essentially all the test sets used after the masking phase are a subset of the train set. After training the network in this initial phase, we then attempt to remove weights that are not necessary to perform certain combinations of commands. Our results clearly demonstrate that such groups of weights exist and that the network performs well without them on all other commands except those removed. Hence, this clearly demonstrates that the learned solution adopts some non-compositional properties, confirming a deeper issue is that hand.

---

> > > > > ### Comment · AnonReviewer1 · 2020-11-19
> > > > > **Slightly misleading**
> > > > >
> > > > > I would be inclined to agree your general argument, if it was not for page 6. There, it says:
> > > > >
> > > > > "It was previously shown that typical NNs generalize poorly on data splits systematically different from the train set [...] : (a) The NN might have learned an algorithm to solve the problem, but the learned representations’ quality is insufficient [...]  (b) Alternatively, the NN might not have learned the correct algorithm [...]"
> > > > > And then continues with " Thus, if the masking process removes any important weights, then the solution is pattern-recognition like instead of being based on reusable rules, confirming explanation (b)"
> > > > >
> > > > > In fact, a similar argument already occurs on page 5. There, the (implicit) argument is that the permutation tasks are similar enough such that the model should be able to reuse if it had learned the right rule (of learning the permutation in the first layer).
> > > > >
> > > > > Given that this argument fills nearly all of page 6, I read it as being the essential argument of the paper. Unless I am severely misunderstanding something, this is an argument as I have described above, with all of my concerns holding against it.
> > > > >
> > > > > If your argument, however, is that "[t]he main argument is that neural networks resist weight sharing to implement shared functionality [...]", then only one part of it is novel (the weight sharing part, given that I hope that we can all agree that NNs resist learning shared functionality in a principled fashion - an argument already made by the original SCAN paper). This is an argument and a line of reasoning in a paper that is somewhat narrow, but that could be convincing by itself, if some analysis as to the how and why of this effect occurring was added. Even then, the "why" has probably to do with the inductive bias of the respective architecture, and the training characteristics of SGD. A CNN for instance would be invariate to shifts of the image. This brings me to my observation on page 5: given that we train the entire network, why would it (in particular an FNN) learn a decomposition that allows it to re-use?
> > > > >
> > > > > However, I may be overly critical, given the other reviewers' take on this paper. I would be particularly interested in what reviewer 4 has to say after having read my concerns above.

---

> > > > > > ### Comment · AnonReviewer4 · 2020-11-20
> > > > > > **Concerns outweighed by benefits**
> > > > > >
> > > > > > I think there is a reasonable level of agreement between myself and Reviewer 1 regarding the weaknesses of the paper in its current form. In particular, we both appear skeptic of conclusions drawn in the paper being sufficiently supported. I have noted this issue in my review and encouraged the authors to more clearly articulate their hypotheses require further evidence, which they agreed was appropriate.
> > > > > >
> > > > > > I also agree with Reviewer 1 that the "A or B, not A, therefore B" line of argumentation presented in Section 4 is somewhat dubious given the extent to which representation and computation are intertwined in NNs. However, I do find the experiments compelling in the sense that they are not conducive of explanation (a) alone. As such, weakening some statements may largely suffice, e.g., changing "confirming explanation (b)" to something like "providing evidence for explanation (b)" or "ruling out explanation (a) alone."
> > > > > >
> > > > > > Despite these issues I would still argue for acceptance of the paper. I see the key contributions being
> > > > > >
> > > > > > 1. A method for identifying submodules within neural networks responsible for particular functionality. In addition to the current suite of experiments, the authors have also proposed an alternate experiment in their response to my initial review which I believe will further justify the approach.
> > > > > > 2. The observation that neural networks tend to feature independent submodules (at least, in the experiments presented here).  I found the mask inversion experiment unexpected in this regard.
> > > > > > 3. Several *hypothesis* regarding why neural networks may fail to reuse functionality.
> > > > > >
> > > > > > Furthermore, while follow-up work may confirm or refute the hypothesis presented in the paper, I believe further exploration of the ideas here would be elucidating and of clear value to the community.

---

> > > > > > > ### Author Response · Authors · 2020-11-23
> > > > > > > **Section 4 improved**
> > > > > > >
> > > > > > > Thank you for clarifying these issues and providing valuable feedback. We believe that the improved draft (which was uploaded minutes ago) should address these concerns.
> > > > > > >
> > > > > > > As to ruling out the representation issues, we believe that we improved Section 4 significantly to clarify what we meant by representation, and also adapted our conclusion to reflect the concerns expressed in your comments.

---

### Official Review · AnonReviewer3 · 2020-10-28
**This work presents a simple yet effective method to investigate the functional modularity of neural networks. Through many carefully designed experiments, the authors found that most of the weights are specialized for a specific function and not shared across different tasks.**

**Rating:** 6
**Confidence:** 3

**Review:**

The paper investigates the functional modularity of neural networks using a simple yet efficient end-to-end method. The paper is well written and clearly articulates a contribution to the literature. The proposed method is intuitive and straightforward. The experimental evidence is provided for both synthetic, language, and image classification tasks. Most of the related works are cited.

Concerns: The biggest concern I had is whether the conclusion reached in the paper is invariant to different neural network architectures, the size of the network, and the complexity of the task. As depicted in Figure 6 (a), there is a huge difference in the relative drop in performance for simple CNN, simple CNN without drouput and ResNet-110. It seems that a larger and more complex network tends to not sharing weights.

Besides, the paper uses accuracy drop after masking the weights as the main metric, which is related to the number of masked weights. It might be better to learn the binary mask for each subtask with a certain accuracy objective (e.g., less than 1-2% lower than the original network) for each subtask and compare the learned mask of different subtasks. In addition, it is useful to see the results on a more complex task such as ImageNet. In ImageNet, there exist many similar subtasks as many categories of the images are actually belong to one big category. I am wondering whether these subtasks can share the weights.
Minor comments: The paper claims the advantage of using Gumbel-Sigmoid than a simple threshold function. However, the state-of-the-art binarized neural networks are trained using a simple sign function with a straight-through estimator. Is there a significant difference (e.g., the stability of the training) in training the binary mask with the Gumbel-Sigmoid and threshold function?

Reasons for score: I vote for accepting. I like the finding that most neural networks have non-overlapped functional modules. However, I still have some concerns about the generality of the conclusion, and also the number of shared weights is not calculated with respect to a uniform accuracy requirement. I would consider raising my score if the authors can address my concerns.

---

> ### Author Response · Authors · 2020-11-17
> **Response to reviewer 3 with comments on improvements**
>
> Thank you for taking the time to review our work in detail and provide valuable feedback. We’re glad that you were able to appreciate the effectiveness of the proposed method as well as our findings.  Below are the individual responses to your specific queries and comments.
>
> > “The biggest concern I had is whether the conclusion reached in the paper is invariant to different neural network architectures, the size of the network, and the complexity of the task.”
>
> We understand your concern. When empirically studying properties of neural networks there will always remain the question of whether they will hold on yet another dataset or using yet another architecture. Unfortunately, reality is that we can only test so many things. In the current submission we present consistent observations across what we believe to be a wide variety of neural networks: FNNs, LSTMS, CNNs, ResNets, Transformers, and data sets: synthetic data sets, datasets for image classification, algorithmic language tasks and solving math problems, which makes it reasonable to assume that similar behavior can be observed in (slightly) different settings. Moreover, the considered architectures contain standardized architectures that are popular across the deep learning literature, and the data sets include several standard benchmarks. Therefore, although we can not with absolute certainty say that these results will extend to other data sets (or perhaps in other domains) or when considering certain custom architectures, we strongly believe that the insights presented in this work will be of use to a significant fraction of the neural networks community. We appreciate that you voted for acceptance even given this concern.
>
> > “It seems that a larger and more complex network tends to not sharing weights.”
>
> Larger networks are certainly more susceptible to not sharing weights. This can also be seen in Figure 9. As the network size shrinks, the network must share weights in order to have enough capacity for solving the task. However, note that sharing is not happening as a result of performing similar functionality as is evident from the findings in our paper. Moreover, in most cases, overparametrized networks tend to work better. For the SCAN and Math experiments, we used the default architecture with default hyperparameters of the original papers. The simple CNN is a reduced version of a popular architectural choice used for classifying MNIST. We emphasize that although the amount of sharing varies, our goal was not to show what fraction is _generally_ shared. Rather we demonstrated that the networks are biased against sharing based on _functionality_, which is consistently observed. We will improve our discussion of these results in the revision to better clarify this.
>
> > “Besides, the paper uses accuracy drop after masking the weights as the main metric, which is related to the number of masked weights”
>
> It is indeed related, but not (directly) caused by it. Weight removal is the result of them not being needed to perform the target task.
>
> Our goal has always been to measure modularity in a way that depends least on the exact amount of weight sharing. For example, this quantity may cause problems when encountering alternative weight configurations, especially when used with dropout: Some weights might be marginally more important and kept in some instances and not in the others as a result of the masking process’s stochasticity. Further, for many tasks it is unclear what a good amount of weight sharing is (see also our reply above to R3 regarding this). In contrast, the desired behavior when applying different masks is more easily understood. This led us to use accuracy for our experiments, since in this case if the experiments are correctly set up, the precise amount of sharing is irrelevant. The effects on accuracy are directly measurable and interpretable.
>
> We hope that this explanation helps motivate our design choices regarding the considered metric. Should you have additional concerns, then please let us know so we can clarify further.
>
> > “It is useful to see the results on a more complex task such as ImageNet”
>
> We briefly mentioned in the paper that we don’t expect the CNN experiments to directly transfer to more complex datasets with a higher number of classes. However, the experiment you propose sounds very interesting. We are currently working on conducting such an experiment with bigger category groups removed initially (for example remove all dogs). However, we note that since Imagenet is very expensive to train on, we might not be able to finish these experiments in the rebuttal period.

---

> > ### Author Response · Authors · 2020-11-17
> > **Response to reviewer 3 with comments on improvements, part 2**
> >
> >
> > > “Is there a significant difference (e.g., the stability of the training) in training the binary mask with the Gumbel-Sigmoid and threshold function?”
> >
> > Interesting question. We don’t have an answer yet, but are currently running experiments with deterministic binary masks on the diagnostic tasks (section 3.1 and 3.2) in order to compare the results with the Gumbel-sigmoid based ones. We will notify you once the submission has been updated to include these results.
> >
> > > “I still have some concerns about the generality of the conclusion, and also the number of shared weights is not calculated with respect to a uniform accuracy requirement.”
> >
> > By “uniform accuracy requirement” do you mean that not all subtasks have the same accuracy? For all subtasks in all experiments, they are trained to have only a marginal accuracy difference compared to the original task (within a few percent). So in this sense, it meets the uniform accuracy requirement. Moreover, only two experiments are based on conclusions about weight sharing: the addition/multiplication and the transfer learning experiments. But for those, the shown effect is so extreme that it cannot be because of noise or alternative weight configurations.

---

> > > ### Author Response · Authors · 2020-11-23
> > > **Update on deterministic masks**
> > >
> > > In addition to the general updates of the paper, we would like to provide you with an update on the deterministic mask experiments.
> > >
> > > We worked on reproducing the double-addition experiments (Section 3.2) with deterministic masks. We binarized by thresholding and made the masks differentiable by the straight-through estimator. We consider this a good benchmark because the stochastic method can discover a mutually exclusive partitioning, shown by the mask inversion experiments. We were unable to recover this partitioning with the deterministic masks. The performance of both pairs decreased with mask inversion. We tried tuning the mask learning rate and regularization strength without success. These experiments are not exhaustive given the time limit, but it suggests that the stochastic masks might have superior discriminative power. One reason might be that the stochasticity helps “explore” the possible weight configurations better than the deterministic method.

---

### Official Review · AnonReviewer2 · 2020-10-28
**Interesting investigation on an important problem. Some approach can be defined more rigorously.**

**Rating:** 6
**Confidence:** 4

**Review:**

The paper presents an empirical study of whether modularity can emerge within neural networks. It starts by proposing a novel definition of modularity that identifies modules by their functionality. To discover the module that implements a specific target functionality, the paper proposes to first pretrain the full network on the original task, then freeze the pretrained weights, and train a binary mask for each weight using Gumbel-Sigmoid. The training objective for the masks is given by the target functionality (e.g., a subtask of the original task), plus some sparsity regularization. The paper then investigates the discovered modules in terms of specialization, reusability, and compositionality. The main findings are: (1) Neural nets tend to satisfy specialization but not reusability; (2) Weight sharing between modules tends to be affected more by whether I/O are shared than by task similarity, and there tends to be less sharing in larger networks; (3) When trained on algorithmic tasks, neural nets fail to learn compositional rules, and thus generalize poorly; (4) CNNs trained for image classification contain class-specific, non-shared weights in the feature detectors.

Pros
- The experiments are comprehensive, covering many neural net architectures and datasets. The results seem consistent across these architectures and datasets. Also, source code is provided, and the supplementary material provides sufficient details to reproduce the results.

- The paper investigates natural emergence of functional modules, which is a novel perspective on modularity.

Cons
- Some concepts are not rigorously defined. For example, in P_{specialize} and P_{reuse}, when should two modules be considered the same? The paper seems to define a module as a subset of weights. Does this imply that two different subsets (potentially differ by only one element) will correspond to two different modules? If this is the case, then P_{reuse} is extremely hard to achieve, and it can never be true if the input/output neurons of two modules are different (i.e., separate I/O considered in Section 3.2). Hence, the experiments in Section 3.2 seem meaningless. One may argue that the weights in the input/output layer should be excluded from the module. But then the functionality of the module is not well defined. Another confusion I had is: What can be a target functionality and what cannot? Take the addition/multiplication task as an example. The task is basically to learn the function f(a,b,s), where a and b are two numbers, and s is a switch indicating whether addition or multiplication should be performed for a and b. The paper suggests f(a,b,s=+) and f(a,b,s=*) as two target functionalities, which are the original function restricted to two subsets of the input space. Is it reasonable to consider f(a=99,b,s) a target functionality? Would the result be different? Also, all other experiments in the paper seem to construct target functionalities by similarly restricting the original functions. Are there other ways to define target functionalities?

I am somewhat skeptical about investigating properties like compositionality and generalization after network pruning. It is well known that training performance does not reflect these properties. So even if training performance only slightly drops after network pruning, the pruned network may lose some properties that the full network has.

Minor Comments
- It is mentioned that separate I/O biases the network at initialization time to not share weights, but to me it seems to only bias the input/output layers rather than the hidden layers. It would be interesting to see if shared I/O will also lead to unshared hidden weights.

- Fig. 13 did not mention which LSTM variant was used.

- In Section 3.3, why is increased sharing in the first layer undesirable? It might not be possible to undo the permutation with only one layer.

---

> ### Author Response · Authors · 2020-11-17
> **Response to reviewer 2 with comments on improvements**
>
> Thank you for taking the time to review our work in detail and provide valuable feedback. We’re glad that you were able to appreciate the novelty of our perspective as well as the thoroughness of our experimental study.
>
> Before we dive into individual responses to your specific queries and comments, we would first like to briefly comment on point (3) of your summary. In particular, the novelty of our work is not merely in showing that neural networks generalize poorly on algorithmic tasks (which was previously shown in several other works [Lake & Baroni, 2018; Saxton et al., 2019]), but rather that the learned solution on these tasks is non-compositional, even for the training set. This provides counter evidence to the hypothesis that neural networks are unable to find an ‘analogical mapping’ between the training and validation set.
>
> > "Some concepts are not rigorously defined. For example, in $P_{\text{specialize}}$ and $P_{\text{reuse}}$, when should two modules be considered the same? The paper seems to define a module as a subset of weights. Does this imply that two different subsets (potentially differ by only one element) will correspond to two different modules?"
>
> This is an interesting question and indeed it would be problematic if two modules would be considered as “entirely different” were they to vary only by a single element. Therefore, in this work, we have treated $P_{\text{specialize}}$ and $P_{\text{reuse}}$ as a continuous quantity ranging from 0 (no sharing) to 1 (completing sharing), which accordingly allows us to evaluate the _degree_ to which modularity emerges. We will update the next revision of the current submission to better emphasize this distinction.
>
> Further, and related to this, we note that throughout most of the work we do not rely on these quantities directly. Rather, we measure the change in performance when certain masks are applied, which is well defined and yields what we believe to be an easy to interpret metric. In this way, we decouple our findings from the specific amount of sharing that is observed, which is a quantity that is hard to interpret (i.e. how much sharing is desirable on say CIFAR10?). On the contrary, it is clear how the performance should change as a result of applying different masks (e.g if the same functionality is used twice, their weight should not be mutually exclusive, as we demonstrated in section 3.2). We will further clarify our motivation for this metric in the paper.
>
> > “Hence, the experiments in Section 3.2 seem meaningless”
>
> Hopefully from our previous comment it is now clear that the experiments in Section 3.2 are highly meaningful. To summarize, we show that when performing two addition operations ($x=a+b$ and $y=c+d$), one becomes completely independent of the other. This is achieved by training a mask to select the subset of the network responsible for $x=a+b$, which performs well for $x$, but the performance of $y$ is low. However, we noted that this behavior could also be observed if just a subset of weights is unshared. Hence, we also take the inverse of the mask trained on $x$ (which now excludes all the weights needed for $x=a+b$), and demonstrate how the resulting network performs well on $y$ (while performing poorly on $x$). This clearly demonstrates that two independent modules are learned: they correspond to two mutually exclusive sets of weights, which is a non-trivial finding.
>
> > “One may argue that the weights in the input/output layer should be excluded from the module”
>
> Ideally, one would want to exclude the input/output connections from the analysis. However, they are already performing a transformation of data, which makes it impossible to do so without architectural modifications (and in that case the functionality of a module is not well defined as you noted). We combat this issue in two different ways: either we use shared input and output (for the addition/multiplication task, SCAN, Math dataset, CNN), or we made sure that the I/O sharing does not affect the results (for double addition, in Section 3.2, we find two mutually exclusive sets of weights, each one capable of performing one, and only one of the two instances of addition).

---

> > ### Author Response · Authors · 2020-11-17
> > **Response to reviewer 2 with comments on improvements, part 2**
> >
> > > “What can be a target functionality and what cannot?” “Is it reasonable to consider f(a=99,b,s) a target functionality? Would the result be different?”
> >
> > In principle any operation that the network is able perform can be used as a target functionality. This includes partitioning of the dataset, or even novel tasks if the network can generalize. The resulting masks will highlight which weights are responsible for it. For example, if we consider f(a=99,b,s) as a target functionality, it could give us insight as to how certain operations are performed internally. Specifically, we would gain insights on how the number 99 is handled, when used as the first operand. For example, it could be interesting to compare this with other numbers, or when using 99 as the second operand to see how much the resulting masks overlap.
> >
> > We argue that this flexibility is a strength of our method as the notion of a ‘desirable’ module is generally also task-dependent. This is reflected in our experiments, where we considered target functionalities that reflect properties that are generally preferred in a network capable of compositional generalization. For example, it is reasonable that the adder should be shared when addition is performed twice. It also is intuitive to use the same weights for computing all possible (valid) combinations of the input tokens on SCAN with the same weights (given that all possible rules are covered in all splits). We will update the current draft to include an improved discussion of choosing potential target functions and any limitations that may be encountered.
> >
> > > “I am somewhat skeptical about investigating properties like compositionality and generalization after network pruning. It is well known that training performance does not reflect these properties”
> >
> > We would like to point out that we never measure generalization directly: we always measure performance on the subset of the training set used for training the _weights_, while training the masks on the complementary subset. From the perspective of the weights, this is a measurement on their train set. Thus, the generalization behavior of the original network should not affect our measurement.
> >
> > Notice however that these measurements are still informative: if certain functionalities are not shared even on the train set, they will certainly not work on novel combinations in the validation set, and thus not perform well overall. For example, if there are multiple different subsets of weights responsible to produce the JUMP output in SCAN (each responsible for different situations) we can not expect the network to work in a novel situation given by a novel composition not encountered in the train set.
> >
> > > “It would be interesting to see if shared I/O will also lead to unshared hidden weights.”
> >
> > In cases where we have used unshared I/O, this is the only difference between tasks. Meaning that if we would use shared I/O as you suggest, there would no longer be two different tasks, but rather we would train a single network on the same. Naturally this would result in all weights always being shared in this case.
> >
> > > “Fig. 13 did not mention which LSTM variant was used.”
> >
> > Thank you for pointing this out. We will update this in the revision.
> >
> > > “In Section 3.3, why is increased sharing in the first layer undesirable? It might not be possible to undo the permutation with only one layer.”
> >
> > We note that for Permuted MNIST, undoing the permutation with just the first layer is always possible when the layer is re-learned, as is considered here (the weights have to be permuted similarly to the input permutation). However, we agree that certain permutations of weights could be impossible to learn and result in weights being shared, when the corresponding weight in the first layer is already used and frozen by the previous permutation. On the other hand, considering how sparse the input layer is, this is highly unlikely (very few pixels are important due to the simplicity of MNIST). Nevertheless, following your comment and a similar comment by R4 we are conducting a new experiment where we explicitly re-initialize the first layer to prevent this effect from happening. In that case, with absolute certainty the permutation can always be undone. We will let you know once the results are added to the revision.

---

### Author Response · Authors · 2020-11-23
**Changes to the paper addressing the reviewers comments and suggestions**

We have uploaded a new version of the submission, which we believe addresses almost all of the comments and suggestions made by the reviewers, including several new experiments and evaluations.

Perhaps most importantly, there was some controversy regarding some of the claims made in Section 4 (on analyzing systematic generalization on algorithmic tasks). We acknowledge that our observations are insufficient to provide a definitive answer as to why neural networks fail to generalize systematically on the SCAN and Math tasks, which was an oversight on our part. We very much appreciate the comments by R1 and R4 regarding this matter and we have adjusted the text accordingly. In particular, we now focus on ruling out explanation (a) as being the _only_ reason for the observed behavior and how we believe that (b) is a logical _hypothesis_ that follows from this. This reflects that there may be other reasons for the observed behavior that are not fully captured by explanation (b). We have also refined our explanation of hypothesis (a) and (b) as to not make controversial statements.

More specifically, we have made the following changes to the submission:

- (R1) The plots from Appendix C.4 regarding the double addition experiments (which were referenced in the second paragraph on page 5) are now part of the main text. The results for LSTM with separate inputs at each timestep were incorporated in Table 1 (bottom two rows). Similarly, Figure 12 is now Figure 2 in the main text.
- (R1) We have clarified the distinction between the ability to reuse modules and the ability to compose them. We have also clarified how we distinguish between modularization and specialization in the context of the two desirable properties that we study.
- (R1) We now explicitly mention how prior approaches to continual learning and transfer learning freeze used weights via masking to avoid suggesting that our approach is novel in this regard.
-(R1) We have added a discussion of Bengio et al. (2015) in the related work section.
- (R2) We have clarified that in this work $P_{\text{specialize}}$ and $P_{\text{reuse}}$ are treated as continuous quantities, which lets us draw valid conclusions.
- (R2) We have added a discussion of how the target function can be chosen in addition to clarifying why we use change in accuracy after masking as the main metric. This includes a discussion of how results should be interpreted in various scenarios.
- (R2/R3) We have also clarified our motivation for measuring the change in performance after applying a particular mask as the main metric, i.e. how this lets us draw conclusions without assuming prior knowledge about the precise amount of weight sharing (which is typically unknown). This was additionally emphasized when motivating our analysis and how results can be interpreted in Section 2.
- (R4) We have changed how we report results on the double-addition and the addition/multiplication tasks. We now report IoU as you suggested as well as what we call IoMin, which explicitly measures whether the set of weights for one task is a subset of the other (which is not captured by IoU). The changes that were made include a new discussion of these metrics and updated plots.
- (R4) We have replaced Figures 2 and Figure 14 with a single table (in the main text) that now reports these results more concisely.
- (R4) We have clarified why it is important to train the mask and the weights together on the transfer learning experiments, i.e. to be able to bias the network towards weight sharing.
- (R4/R2) We reran the transfer learning experiment, but now with the modification that the first layer is always reset. In this way we ensure that it is always possible to undo the permutation. We now also report the proportion of weights of a task shared with any of the previous tasks. Our findings in Figure 3 are similar to before in that we are unable to observe significant weight sharing, except when capacity becomes an issue. Interestingly, however, we were able to observe different behavior when explicitly biasing the network towards sharing following these changes. We now find that significantly biasing the network towards sharing indeed lets later layers become shared.

---

> ### Author Response · Authors · 2020-11-23
> **Changes to the paper addressing the reviewers comments and suggestions, part 2**
>
> - (R4/R1) We have weakened several of our prior statements regarding the nature of the solution learned by the RNN and Transformer on SCAN (and on Math by extension) and the explicit need for variable binding. While our experiments are able to rule out explanation (a) as the _only_ problem, we acknowledge that there may be other reasons for the observed behavior that are not fully captured by explanation (b).We now mostly emphasize how we provide strong evidence against (a) being the only issue and how we believe that (b) is a logical hypothesis that follows from this. We also emphasize that it is a hypothesis that proper variable binding plays a crucial role in a generalizable solution for solving such reasoning tasks.
> - (R4) Similarly in the related work section, we have toned down claims regarding the underlying reason for the observed behavior as was reported in Section 5.
> - (R4) We have conducted an experiment in appendix B where we study the effect of masking on the performed computation. We report the drop in performance for cases where the first half or the second half of the network is masked across a variety of networks and datasets. Generally we find that masking barely impacts performance in both of these cases, providing further evidence for the reported observations. The only exception was for the large FNN, which we believe is due to it being overparameterized. An additional experiment using a smaller FNN appears to confirm this.
> - (R4) We have clarified that node-level may not always be enough to draw meaningful conclusions as a way of communicating that they are not strictly the wrong approach in all cases.
> - (R4) We have added the sanity check you suggested (Appendix C.1) on the double-addition task, where we manually edited the input/output weight matrices to reuse the hidden layers. It can be seen that our method accurately discovers that the hidden layers are shared in this case, further validating the proposed masking mechanism.
>
> We have also made several general changes
>
> - We have added additional analysis regarding the inverted mask experiment on the double-addition task, where we measure the effect of only applying the inverse mask to the hidden layers (and using the mask for the full task on the input/output layers). Our findings are similar to the results reported previously. Further, we considered the effect of not masking the input/output layers, which yields non-trivial observations and calls for further research.
> - We noticed that the y-axis on some of the figures did not always extend to the same range, where this could be reasonably expected. Hence, we have updated the visualizations to use the same range where this is appropriate.
>
> Finally, although we would have liked to incorporate results on ImageNet (following a suggestion from R2), we will most likely not be able to complete these experiments in time. Nonetheless we seek to add the experiment that was suggested in our reply in a future version of this work.

---

### Decision · Program_Chairs · 2021-01-07
**Final Decision**

**Decision:**

Accept (Poster)

**Comment:**

This is a paper that is actively discussed.  The general sentiment is that this paper aims to address an important set of questions. While the technique could be improved with more novelty, the empirical study is extensive. The concerns are about how to interpret the results, or rather whether the empirical evidence fully supports the the claim/hypothesis.  After discussion and rebuttal, the reviewers improved their scores (and one reviewer remained at "weaker marginally above threshold").

The AC read the paper and the discussion. One value the AC sees that the discussion threads between the authors and the reviewers provide a significant amount of scientific value -- the questions to be answered are hard and might indeed require further refinements in framing and conceptualization, better techniques,  strong power in experimental designs to rule exclusively various hypotheses.  Thus, the AC recommends acceptance.